# Cystic proliferation of germline stem cells is necessary to reproductive success and normal mating behavior in medaka

**Luisa F Arias Padilla[1], Diana C Castañeda-Cortés[2], Ivana F Rosa[2], Omar D Moreno Acosta[3], Ricardo S Hattori[2], Rafael H Nóbrega[1], Juan I Fernandino[1]***

[1]Instituto Tecnológico de Chascomús, INTECH (CONICET-UNSAM), Chascomús, Argentina; [2]Reproductive and Molecular Biology Group, Department of Structural and Functional Biology, Institute of Biosciences, São Paulo State University (UNESP), Botucatu, Brazil; [3]Salmonid Experimental Station at Campos do Jordão, UPD-CJ, Sao Paulo Fisheries Institute (APTA/SAA), Campos do Jordao, Brazil

**Abstract** The production of an adequate number of gametes is necessary for normal reproduction, for which the regulation of proliferation from early gonadal development to adulthood is key in both sexes. Cystic proliferation of germline stem cells is an especially important step prior to the beginning of meiosis; however, the molecular regulators of this proliferation remain elusive in vertebrates. Here, we report that *ndrg1b* is an important regulator of cystic proliferation in medaka. We generated mutants of *ndrg1b* that led to a disruption of cystic proliferation of germ cells. This loss of cystic proliferation was observed from embryogenic to adult stages, impacting the success of gamete production and reproductive parameters such as spawning and fertilization. Interestingly, the depletion of cystic proliferation also impacted male sexual behavior, with a decrease of mating vigor. These data illustrate why it is also necessary to consider gamete production capacity in order to analyze reproductive behavior.

**\*For correspondence:**
fernandino@intech.gov.ar

**Competing interests:** The authors declare that no competing interests exist.

## Introduction

Cell division and differentiation are well-controlled processes for the development of a functional organ in multicellular organisms. In gonads, these processes are also essential to guarantee the correct production of gametes necessary for successful reproduction (*Lombardi, 1998*). Vertebrate gonads acquire sex-specific features during gonadogenesis, an exceptional, complex, multistep process that includes migration of primordial germ cells (PGCs) to reach the gonadal primordium, proliferation, differentiation, and meiosis (*DeFalco and Capel, 2009*; *Richardson and Lehmann, 2010*). Defects in any of these steps may lead to sex abnormalities including infertility and sex reversal.

Proliferation of embryonic germline stem cells (EGSCs, the name applied to PGCs after they reach the gonadal primordium) gives rise to a cohort of germ cells that will ultimately form the primary population of germ cells in the gonads of both sexes (*Tanaka, 2019*). This type of slow, intermittent division, also called type I proliferation, is similar to the self-renewal of germline stem cells in the adult ovary or testis (*Chen and Liu, 2015*; *Nakamura et al., 2010*), and is responsible for ensuring the production of the correct number of germline stem cells that will initiate gametogenesis (*Tanaka, 2019*). In addition, the hyperproliferation of mitotically active germ cells in *hotei* medaka, a mutant fish for anti-Müllerian hormone receptor II (*amhrII*), induces 50% male-to-female sex reversal and loss of reproductive success in adult fish (*Morinaga et al., 2007*; *Nakamura et al., 2012*). On the contrary, lack of this proliferation of EGSCs during the embryo stage induces the opposite sex

reversal in zebrafish and medaka (*Kurokawa et al., 2007*; *Rodríguez-Marí et al., 2010*), showing that this initial proliferation can be also involved in gonadal sex fate.

A second EGSC proliferation occurs prior to meiosis, following a special program of synchronous, successive and incomplete mitosis, generating interconnected daughter cells known as cysts or nests, which are surrounded by somatic cells (*Saito et al., 2007*). Following several rounds of mitosis, the cells enter meiosis to become oocytes in the ovary or spermatocytes in the testis (*Lei and Spradling, 2013b*; *Matova and Cooley, 2001*; *Tanaka, 2016*). Cystic division is a highly conserved mechanism that precedes gametogenesis in both invertebrates and vertebrates (*Amini et al., 2014*; *Hansen and Schedl, 2006*; *Hinnant et al., 2017*; *Lei and Spradling, 2013a*; *Quagio-Grassiotto et al., 2011*). It has been well established in vertebrates that female germ cells enter meiosis earlier, while male germ cells are arrested and undergo meiosis at the onset of the prepubertal period (*Elkouby and Mullins, 2017*; *Koubova et al., 2006*; *Lei and Spradling, 2013a*; *Pepling, 2006*). In medaka, this type of cystic cell division is known as type II proliferation, by which, in females, the number of EGSCs increases dramatically during embryonic stages; whereas in males, this exponential proliferation to form cysts begins later, in the juvenile stage (*Satoh and Egami, 1972*; *Tanaka, 2019*). Despite the importance of cystic cell proliferation during early life to ensure reproductive success, this process is not fully understood at the molecular level (*Elkouby and Mullins, 2017*).

In the search for genes that regulate EGSCs proliferation, *ndrg1* (n-myc downstream regulated gene 1) is one of the genes that was found to be upregulated in germ cells with mutated *foxl3*, a switch gene involved in the germline sexual fate decision in medaka (*Kikuchi et al., 2019*). Interestingly, *Ndrg1* is involved in the regulation of cell proliferation in different cancers (*Chang et al., 2014*; *McCaig et al., 2011*; *Xi et al., 2017*; *Zhang et al., 2019*). Moreover, this gene has homologs from mammals (*Ndrg1*) to fish (*ndrg1*), *Caenorhabditis elegans* (ZK1073.1) to fruit fly (Dmel\MESK2), suggesting that NDRG1 is highly conserved among species (*Kovacevic et al., 2013*; *Sun et al., 2013*). *Ndrg1* has been demonstrated to inhibit the TGF-β signaling cascade, a very well-known pathway involved in cell proliferation and survival (*Zhang et al., 2017*), regulating the induction of epithelial-to-mesenchymal transition in mammalian cells (*Tojo et al., 2005*; *Chen et al., 2012*; *Jin et al., 2014*), establishing *ndrg1* as a potent inhibitor of proliferation. However, its role in non-cancerous cells is not well established. In this study, we provide clear evidence of the important role of *ndrg1* in the regulation of EGSC cystic proliferation, with subsequent impact on reproductive success in medaka.

## Results

### Expression of *ndrg1* during gonadal development

To understand the role of *ndrg1* in cystic proliferation, the expression pattern of *ndrg1* during gonadal development was first determined. *ndrg1* belongs to the n-myc downstream regulated gene 1 family, and in the medaka genome has two paralogs (*Figure 1A*), as a result of teleost-specific genome duplication (*Kasahara et al., 2007*): *ndrg1a* (ENSORLG00000003558) on chromosome 11 and *ndrg1b* (ENSORLG00000004785) on chromosome 16, with 60% identity. The *ndrg1a* transcripts were not detectable in the medaka gonads during embryo developmental stages by whole-mount RNA in situ hybridization (*Figure 1B*). In contrast, the medaka *ndrg1b* was specifically observed in gonads during embryo developmental stages (*Figure 1C*). The transcript level of *ndrg1b* was quantified by qPCR at stage 35, when female (XX) and male (XY) gonads appear morphologically indistinguishable and germ cells only show type I proliferation. No difference for *ndrg1b* transcript levels was detected between female and male individuals (*Figure 1D*). Interestingly, differences between sexes were observed at stage 39, at 10 and 20 days post hatching (dph) (*Figure 1E–G*). At stage 39 and 10 dph, where female gonads have higher number of cysts compared to male gonads due to type II cystic division, the level of *ndgr1b* transcripts is higher in male than in female gonads (*Figure 1E, F*). The opposite sex transcriptional profile of *ndrg1b* was observed at 20 dph (*Figure 1G*), when type II proliferation begins in male individuals. These results showed that the transcription of *ndrg1b* is inversely related to the cystic proliferation of EGSCs in both sexes.

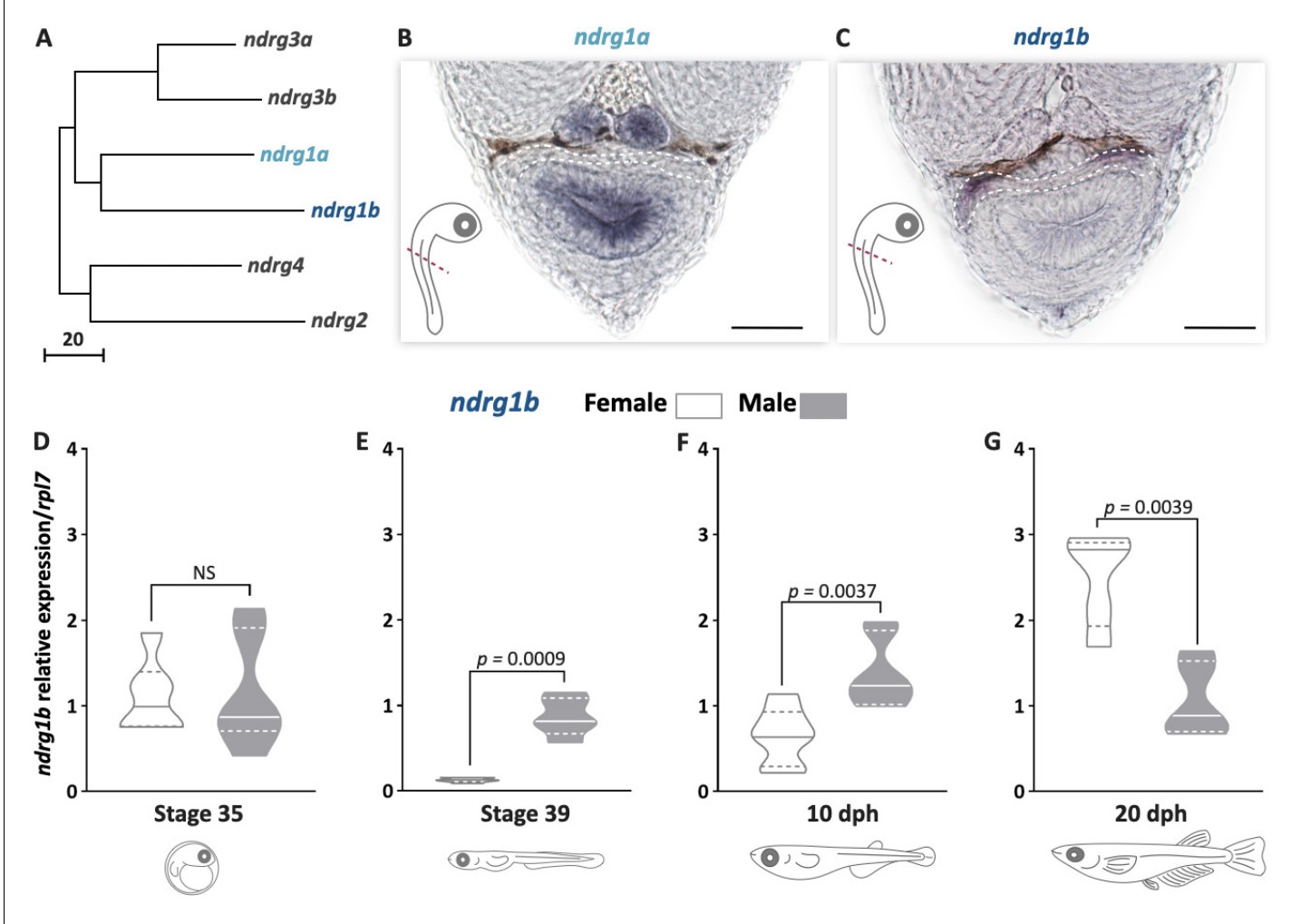

**Figure 1.** *ndrg1b* is down-regulated during cystic proliferation in gonadal development. Phylogenetic tree showing the relationships among the *ndrg* family members in medaka (A), obtained using the neighbor-joining method and a bootstrap test (MEGA 7.0 software). The scale beneath the tree reflects sequence distances. ENSEMBL accession sequences are provided in *Supplementary file 1A*. Transversal sections of gonadal region (red line) in whole embryos at stage 35 to detect *ndrg1a* (B) and *ndrg1b* (C) transcripts using in situ hybridization. Gonadal region is surrounded by white dotted line; scale bar represents 20 µm. Transcript abundance levels of *ndrg1b* in different stages of gonadal development: 35 (D), 39 (E), 10 days post hatching (dph) (F), and 20 dph (G). Quantification was performed using the $2^{-\Delta\Delta Ct}$ method and *ndrg1b* values were normalized to *rpl7*. Genotypic sex was determined by the presence/absence of the *dmy* gene; female (XX) and male (XY) are represented by empty bars or full bars, respectively. n = 5–6 pools per sex in stages 35 and 39, and n = 6–7 individuals per sex in 10 and 20 dph. p-Values are indicated when transcript abundance between sexes at the same developmental stage differ significantly (p<0.05). NS: not statistically significant. Relative gene expression levels were compared as described by *Pfaffl, 2001*.

The online version of this article includes the following figure supplement(s) for figure 1:

**Figure supplement 1.** TGF-β is not involved in the cystic proliferation during early gonadal development.

## TGF-β pathway during proliferation and its relation with *ndrg1b*

Due to the key role of the TGF-β superfamily in the regulation of type I proliferation in medaka (*Imai et al., 2015*; *Nakamura et al., 2012*) and the effect of *ndrg1b* on that proliferation, we decided to analyze whether TGF-β signaling is involved in the regulation of cystic proliferation (*Figure 1—figure supplement 1B*). Embryos of both sexes were exposed to A83-01 (TGF-β inhibitor), from stage 35 to 39 (*Figure 1—figure supplement 1A*). When total EGSCs were quantified, in exposed females no difference was observed, but an increase in EGSCs in exposed males was observed compared to control males (*Figure 1—figure supplement 1C*). Additionally, no difference in *ndrg1b* mRNA levels was detected between control and treatment for female individuals

(*Figure 1—figure supplement 1D*). In contrast, male individuals showed increased levels of *ndrg1b* following TGF-β inhibitor treatment when compared to control (*Figure 1—figure supplement 1D*), establishing that there is no negative feedback from *ndrg1b* via TGF-β. Overall, these results suggest that the TGF-β pathway is not involved in type II proliferation in medaka.

## Mutation of *ndrg1b* affects early gonadal development in both sexes

To explore the role of *ndrg1b* during gonadal development and cystic proliferation, firstly *ndrg1b* biallelic mutants (indels in F0 individuals genome created by injecting Cas9 and sgRNA) were generated using CRISPR/Cas9 technology. The mutagenesis efficiency for each sgRNA was analyzed by a heteroduplex mobility assay (HMA; *Figure 2—figure supplement 1B, Figure 2—figure supplement 2B*; *Ota et al., 2013*), which reached 96.6% for sg1_*ndrg1b* and 80% for sg2_*ndrg1b* (*Figure 2—figure supplement 1B, Figure 2—figure supplement 2B*). These results led us to select sg1_*ndrg1b* for subsequent analysis. In addition, biallelic mutants were mated with wild-type (wt) fish to generate F1 and then several fish were sequenced to confirm the presence of the indels (*Figure 2—figure supplement 1C*). The data indicate that most cells contained biallelic indels and, consequently, loss of function in *ndrg1b* mutants. None of the embryos analyzed had indels at the analyzed off-target sites for the injected sgRNA of biallelic mutant (*Figure 2—figure supplement 1D, E*). In order to address the possible involvement of *ndrg1b* in sex development, biallelic mutants (F0) were analyzed.

Fish having the deletion Δ31 (deletion of 31 bp) previously identified (*Figure 2—figure supplement 1C*) were selected as founder F1 to produce the F2 generation with expected *ndrg1b* knockout (KO) (*Figure 2—figure supplement 3A*). However, the screening of F2 individuals by means of HMA using genomic DNA from a fin clip (*Figure 2—figure supplement 3C, D*) did not reveal homozygous mutants of *ndrg1b* (*Figure 2—figure supplement 3B*).

## Mutation of *ndrg1b* affects early gonadal development in both sexes

To analyze the phenotypic alterations of gonad development in *ndrg1b* mutants, the number of EGSCs was counted using the germ cell marker *Olvas* at stage 39 (*Figure 2A–C*). The total number of EGSCs in male *sgN1b* was unchanged with respect to the male wt individuals. In females, however, the number was significantly reduced in female *sgN1b* compared to the female wt individuals (*Figure 2D*), decreasing to a level similar to male wt. Moreover, when the number of type I and II germ cells in mutants was quantified, only type II cystic proliferation was altered in female mutants compared with wt female individuals (*Figure 2F*). Furthermore, the number of EGSCs inside each cyst in *sgN1b* females was lower than in wt (*Figure 2E*).

We then wondered whether the loss of EGSCs was due to reduced cystic proliferation or to apoptosis in the *ndrg1b* mutants. PCNA immunostaining (*Figure 2B, C*) revealed that the number of proliferating type I and type II germ cells/total number of germ cell at stage 39 did not differ among female *sgN1b* and male gonads, but the number of proliferating type II germ cells was reduced in female *sgN1b* when compared to female wt (*Figure 2G*). In a TUNEL assay, positive cells were not observed in any *sgN1b* or wt individuals (*Figure 2—figure supplement 4A–C*). Hence, female *ndrg1b* mutants have fewer proliferating germ cells at stage 39. These results indicate that type II cystic proliferation is affected in *ndrg1b* mutants. In addition, we performed an extra check of the efficacy of the loss of function of *ndrg1b* with the *sgN1b* guide by analyzing the phenotype of loss of cystic proliferation in individuals injected with a second RNA guide, *sg2_ndrg1b*. We observed the same results as the *sgN1b* biallelic mutants (*Figure 2—figure supplement 2C–E*). These results confirm the *ndrg1b* loss of function observed with the first sgRNA.

The effect of *ndrg1b* mutation was further analyzed during juvenile development in both sexes at 20 dph, which is the time when type II cystic division begins in males. Histological examination showed ovaries with apparently normal development (*Figure 3B*); however, there is a decrease in the number of oogonia and pre-vitellogenic oocytes compared to wt individuals (*Figure 3D, E*). In males, the mutant showed a significant reduction of cysts and germ cells within the cyst in relation to wt males (*Figure 3F, G*). This indicates that germ cell number is affected in *ndrg1b* mutants for both sexes at times that are key to the commitment of type II cyst division.

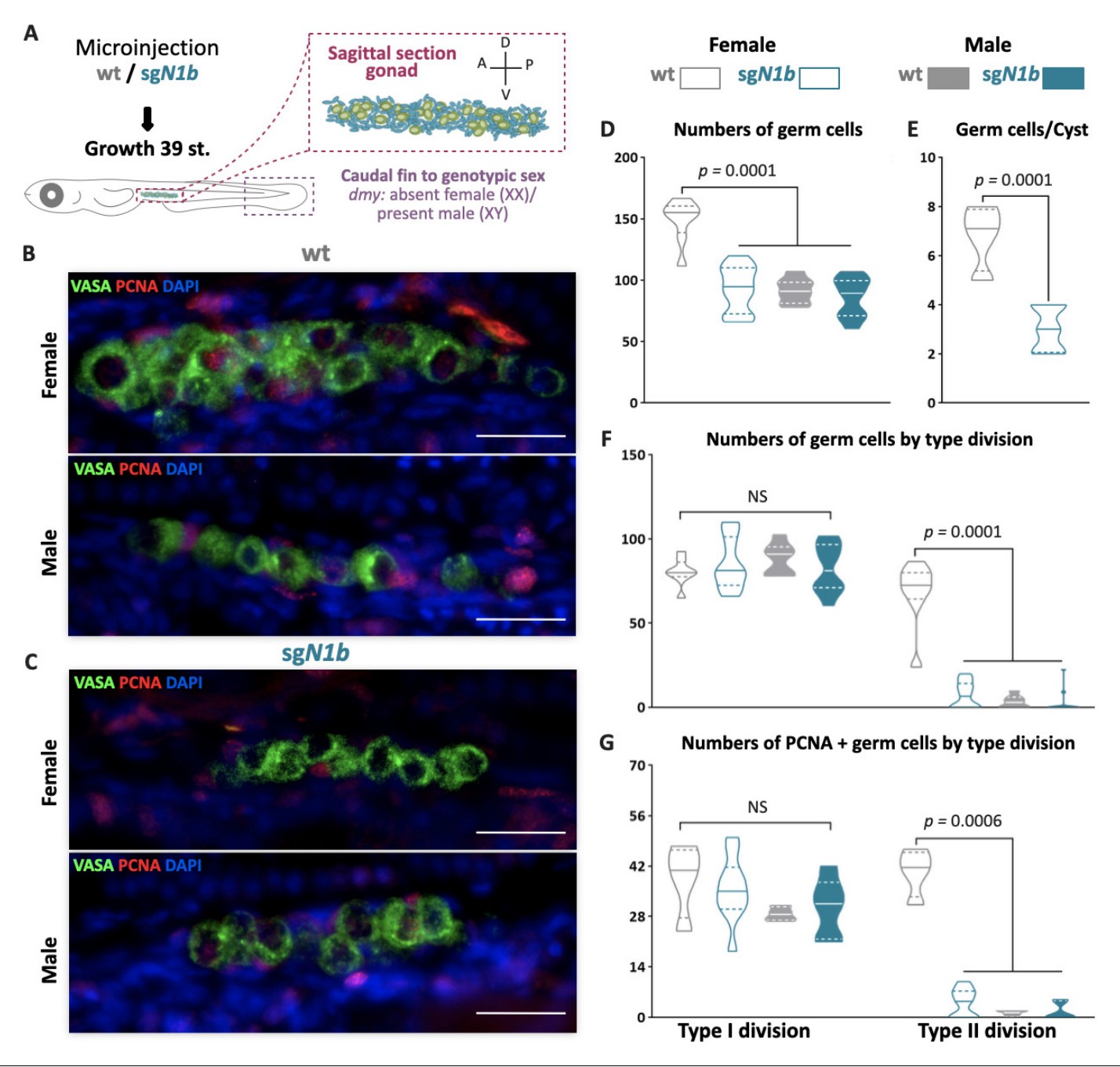

**Figure 2.** The mutation of *ndrg1b* affects gonadal development at stage 39. Schematic representation of the experimental procedure adopted to analyze the effect of *ndrg1b* loss on germ cell number and proliferation (A: anterior, P: posterior, D: dorsal; and V: ventral) (A). Fluorescent images of gonads in female or male embryos; wild-type (wt) = injected with *cas9* (B) and *sgN1b* = injected cas9+sg1_*ndrg1b* (C). Embryonic germline stem cells (EGSCs) were stained with an anti-OLVAS antibody (green), nuclei were stained with DAPI (blue), and proliferating cells with PCNA (: Proliferating cell nuclear antigenDMSO: Dimethyl sulfoxide) antibody (red), scale bar represents 20 μm. Total numbers of EGSCs (D), number of EGSCs per cyst with type II division (E), total numbers of germ cells with type I or type II division (F), and number of PCNA-positive type I and cystic type II germ cells relative to the total number of type I or type II germ cells, respectively (G), in wt female or male embryos (represented by gray empty bars or full bars, respectively) and sg*N1b* female or male embryos (represented by cyan empty bars or full bars, respectively). Vertical bars indicate mean, with its respective standard error of the mean. n = 8 per each wt group and n = 12 per each *sgN1b* group. p-Values are indicated when the number of germ cells between groups differs significantly (p<0.05). NS: not statistically significant. Unpaired Student's *t*-test per E and Tukey's multiple comparisons test per D, F, and G.

The online version of this article includes the following figure supplement(s) for figure 2:

*Figure 2 continued on next page*

*Figure 2 continued*

**Figure supplement 1.** CRISPR/Cas9: sg1_*ndrg1b* design, heteroduplex mobility assay (HMA), efficiency, and potential off-targets.
**Figure supplement 2.** Corroboration of the specificity of CRISPR/Cas9 methodology to mutate *ndrg1b* with a second RNA guide (sg2_ndrg1b).
**Figure supplement 3.** Identification of *ndrg1b KO* fish in an F2 trial.
**Figure supplement 4.** Apoptosis of embryonic germline stem cells (EGSCs) at stage 39.

## Mutation of *ndrg1b* alters gametogenesis and reproductive success of both sexes in adulthood, but not androgen production in males

To evaluate whether impaired type II cystic proliferation can affect gametogenesis and reproductive success, paired adult male and female wt and *ndrg1b* mutants were analyzed for 10 days, and then their body and gonadal morphology was analyzed according to different reproductive parameters (*Figure 4A*). Regarding the daily frequency of spawning, the *ndrg1b* mutant females spawned fewer eggs than wt females (*Figure 4B*), although fertilization rate was normal (*Figure 4C*). Surprisingly, the wt females paired with male *sgN1b* also spawned fewer oocytes (*Figure 4B*), and in *sgN1b*

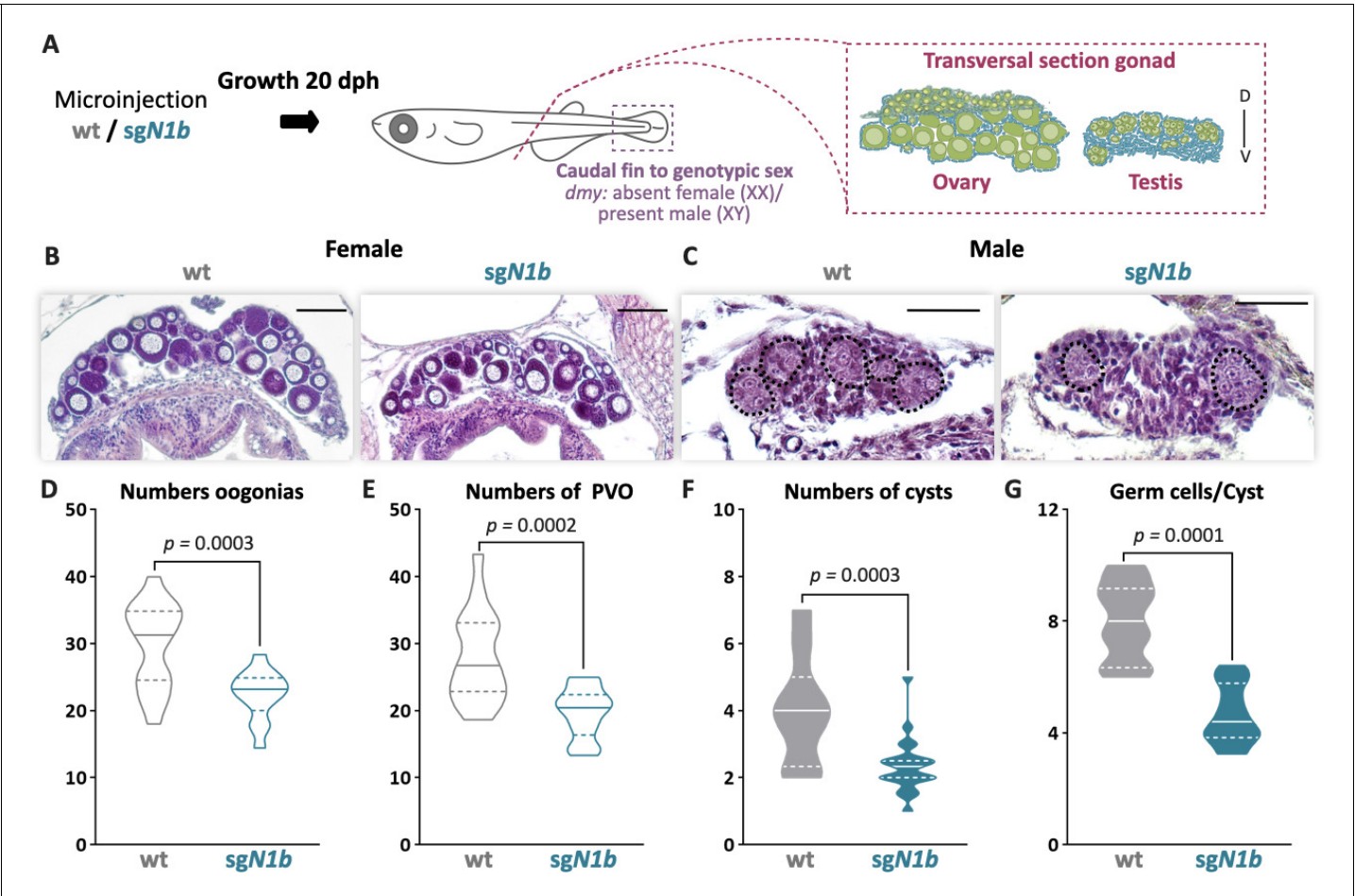

**Figure 3.** The mutation of *ndrg1b* affects gonadal development at 20 days post hatching (dph). Schematic representation of the experimental procedure adopted to analyze the effect of *ndrg1b* mutation on gonadal histology at 20 dph (D: dorsal, V: ventral) (**A**). Histological transverse sections of gonads. Wild-type (wt) = injected with *cas9* or *sgN1b*=injected with cas9+sg1_*ndrg1b* from individuals for XX (**B**) and XY (**C**). In the testis, each cyst of germ cells (spermatogonia-like cells) is encircled by a black dotted line. Scale bars in **B** and **C** represent 50 µm. Total number of oogonia (**D**), number of pre-vitellogenic oocytes (PVO) (**E**), number of spermatogonial cysts (**F**), and number of spermatogonia per cyst (**G**) in wt female or male embryos (represented by gray empty bars or full bars, respectively) and sg*N1b* female or male embryos (represented by cyan empty bars or full bars, respectively). n = 16 per female groups and n = 20 per male groups. p-Values are indicated when the number of germ cells between groups differs significantly (p<0.05). NS: not statistically significant. Unpaired Student's *t*-test.

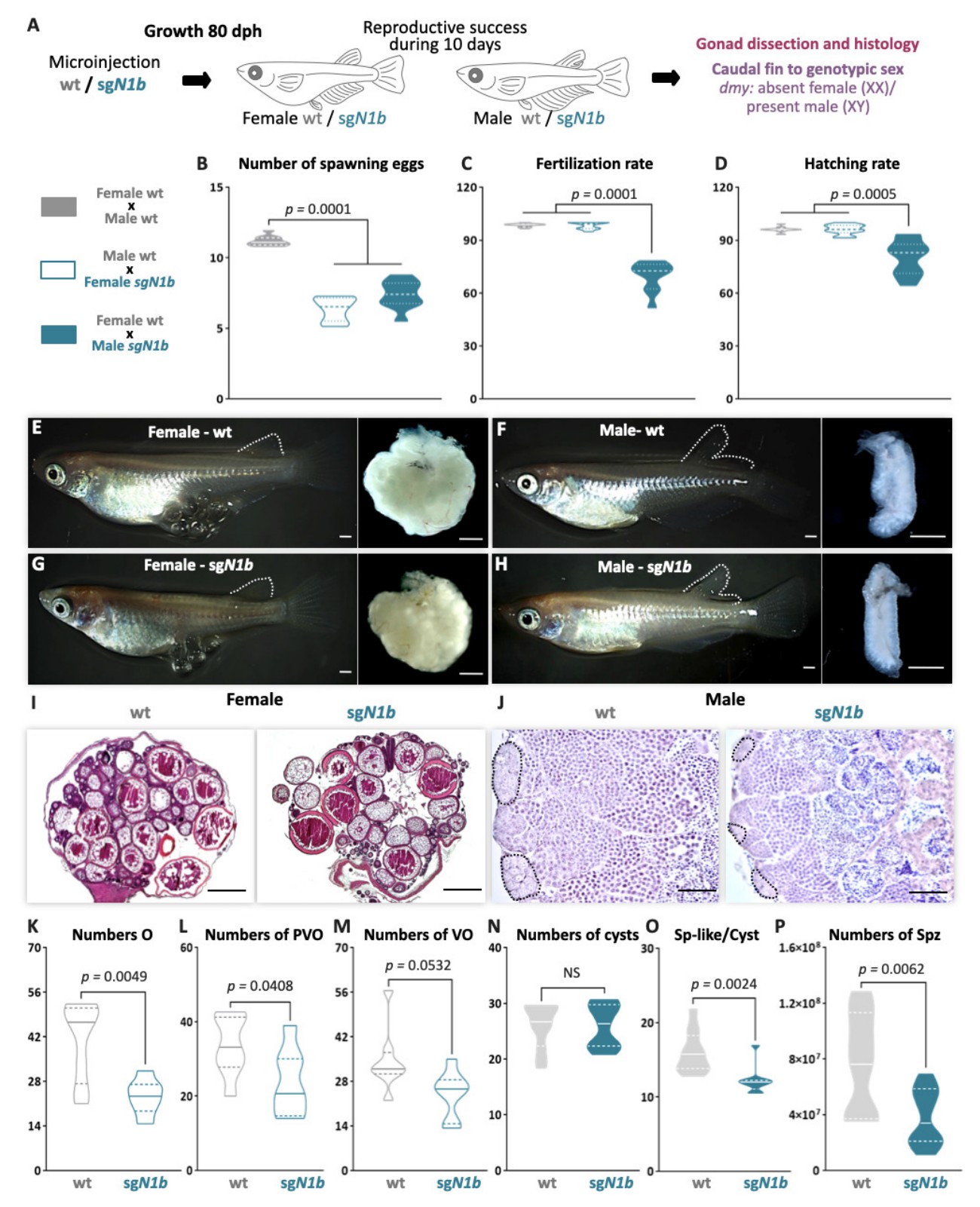

**Figure 4.** Mutation of *ndrg1b* alters the gametogenesis and reproduction success of both sexes in adults. Schematic representation of the experimental procedure adopted to analyze the effect of *ndrg1b* loss of function in the reproductive success at 80 days post hatching (dph) (A). Reproductive success was evaluated by crossing the wild-type (wt) males with *sgN1b* females (cyan full), wt females with *sgN1b* males (cyan empty), and as a control, wt females with wt males (gray). For all crossings, we quantified the total number of eggs spawned (B), percentage of fertilization (C), and

*Figure 4 continued on next page*

*Figure 4 continued*

percentage of hatching (D) of the total eggs spawned. The biallelic mutants of *ndrg1b* did not exhibit morphological sexual differences: adult females with their ovaries, wt (E) and *sgN1b* (G), and adult males with their testis, wt (F) and *sgN1b* (H). The secondary sexual characteristics are indicated by the separation of hindmost rays from other rays in the dorsal fin. Scale bars are 1 mm. Histological transverse sections of wt and *sgN1b* gonads from XX (I) or male (J) individuals. In the testis, each cyst of germ cells (spermatogonia-like cells) is encircled by a black dotted line. Scale bars represent 500 μm (I) and 50 μm (J). Number of oogonia (O) (K), number of pre-vitellogenic oocytes (PVO) (L), number of vitellogenic oocytes (VO) (M), number of spermatogonial cysts (N), number of spermatogonia per cyst (O), and number of spermatozoa (P) from wt female or male individuals (represented by gray empty bars or full bars, respectively) and *sgN1b* female or male individuals (represented by cyan empty bars or full bars, respectively). n = 10 per group. p-Values are indicated when groups differ significantly (p<0.05). NS: not statistically significant. Tukey's multiple comparisons test (B–D). Unpaired Student's *t*-test (K–P).

The online version of this article includes the following figure supplement(s) for figure 4:

**Figure supplement 1.** F1 reproductive success was evaluated by crossing the wild-type (wt) males with F1 females (cyan full), wt females with F1 males (cyan empty), and as a control, wt females with wt males (gray).

males paired with wt females, the fertilization rate declined (*Figure 4C*). Moreover, hatching rate was lower for the progeny of male *sgN1b* and wt female than for the progeny of female *sgN1b* and wt male (*Figure 4D*). Similar results were observed in F1 (heterozygote Δ31), in which the number of spawning eggs was reduced in female and male *ndrg1b$^{+/-}$* (*Figure 4—figure supplement 1A*), the fertilization rate was reduced in males (*Figure 4—figure supplement 1B*), and, interestingly, the hatching rate was sex dimorphic, reduced only in embryo from males compared to the respective control male (*Figure 4—figure supplement 1C*), indicating a need for future studies in this direction.

Medaka adults display easily distinguishable secondary sex characteristics, for example, the hindmost rays are separated from other rays in the dorsal fin in the male, but linked together in female. All *ndrg1b* mutants invariably displayed typical female and male secondary sex characteristics (*Figure 4E–H*). A balanced sex ratio was observed in both *ndrg1b* mutants and wt individuals (*Supplementary file 1B*). Histological examination of females showed ovaries with apparently normal development (*Figure 4I*), but there were significantly fewer oogonia, pre-vitellogenic, and vitellogenic oocytes at the different stages in *ndrg1b* mutants than in wt individuals (*Figure 4K–M*). With respect to the histological examination of testis, all stages of spermatogenesis were observed in *sgN1b* mutants (*Figure 4J*). Spermatogonia are surrounded by Sertoli cells at the distal end of the lobule, which are responsible for constantly supplying new germ cells by proliferation (*Iwai et al., 2006*). When these cells were counted manually, the number of cysts containing spermatogonia remained unchanged (*Figure 4N*); however, the number of spermatogonia within each cyst was lower in *sgN1b* than in wt individuals (*Figure 4O*), and the total number of spermatozoa was lower in *sgN1b* testes than in wt testes (*Figure 4P*). Collectively, these results indicate that alterations of cystic proliferation, with the concomitant lower sperm count and fewer oocytes, have direct influence on reproduction. Interestingly, these alterations do not change the general gonadal morphology or secondary sex characteristics in either sex or the sex ratio.

## Mutation of *ndrg1b* disturbs the mating behavior of males

To explore a possible cause of the lower spawning of control females paired with male *ndrg1b* mutants, our next step explored the sexual behavior of these couples. Fortunately, the sexual behavior of medaka consists of a sequence of actions that are easily quantified (*Hiraki-Kajiyama et al., 2019*; *Ono and Uematsu, 1957*; *Walter and Hamilton, 1970*). Briefly, the sequences begin with the male approaching and following the female closely. The male then performs a courtship display, in which the male swims quickly in a circular pattern in front of the female. If the female is receptive, the male grasps her with his dorsal and anal fins (termed 'wrapping'), they quiver together (termed 'quivering'), and the male makes several gentle taps, like 'convulsions' (termed 'convulsions'), and then oocytes and sperm are released. In general, all couples displayed the usual actions of sexual behavior (*Figure 5B–I*), with differences observed only in the actions related to mating behavior, particularly in the wrapping actions related to the stimulation of the female spawning, the duration of quivering, and number of convulsions, which were significantly fewer in *sgN1b* males paired with wt females than in control couples (*Figure 5H, I*). Collectively, these results indicate that alterations of

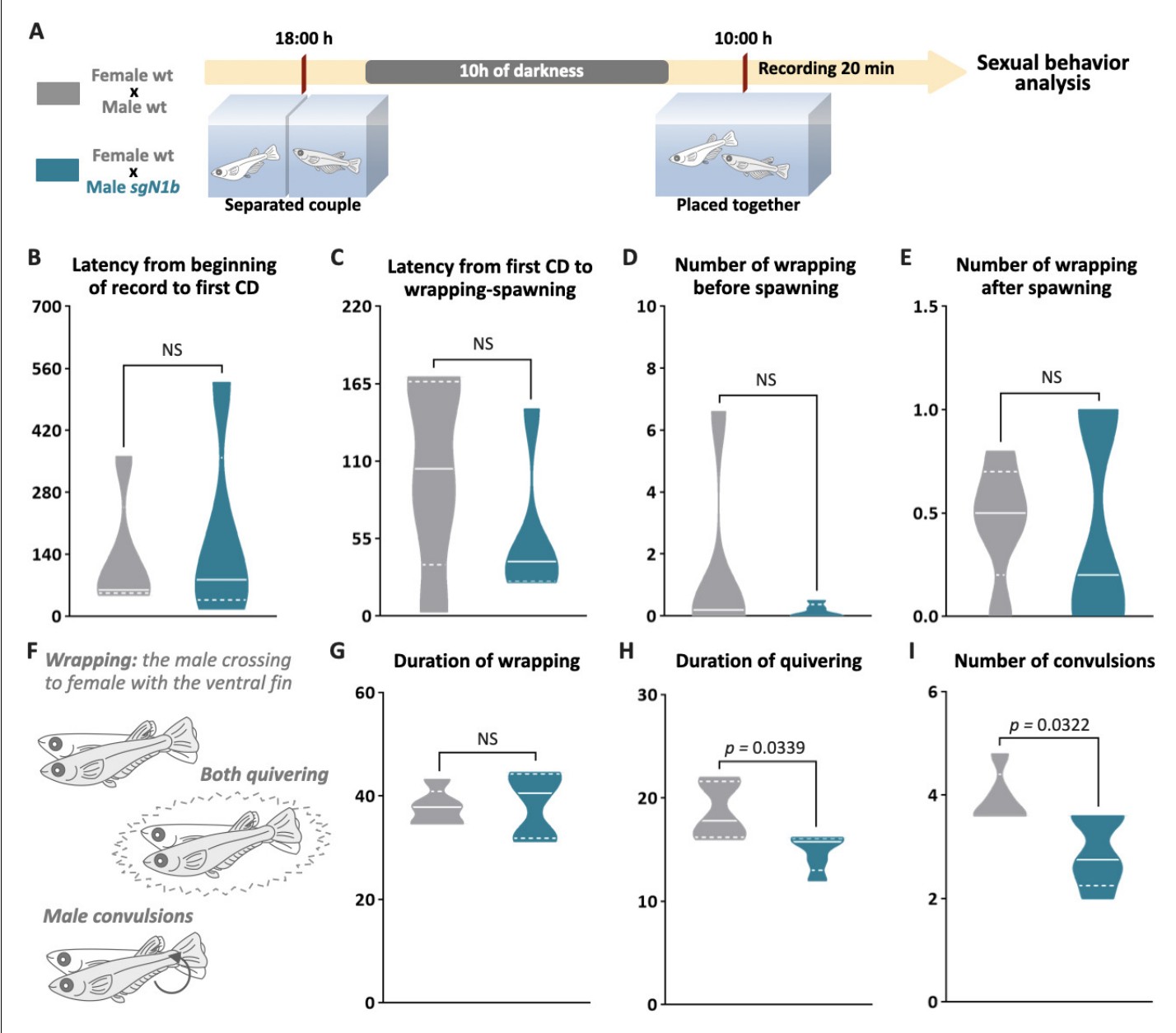

**Figure 5.** Reduction of cystic proliferation in *ndrg1b* mutants decreases male sexual behavior, with a decrease of mating vigor. Schematic representation of the experimental procedure adopted to analyze reproductive behavior for 5 days; five couples per group were analyzed. (A) Couples of wild-type (wt) individuals (gray) or *sgN1b* males with wt females (cyan) were separated in the evening (6–7 PM) the day before the assay using a transparent plastic tank with small holes for water exchange. The following morning, each mating pair was placed together in a single tank and sexual behavior was recorded for 20 min. Each video was analyzed to determinate the latency from beginning of recording to the first courtship display (B), latency from first courtship display to the wrapping that resulted in spawning (C), number of wrappings of 2 s of duration before (D) and after (E) spawning, and specific wrapping actions (F), such as duration of wrapping (G), duration of quivering (H), and number of convulsions (I). Representative videos of wrapping actions in couples of wt female and male (https://www.youtube.com/watch?v=KrI8t90_tMA&feature=youtu.be) and male *sgN1b* with female wt (https://www.youtube.com/watch?v=MCjhYG7lfwM&feature=youtu.be). n = 5 per group. p-Values are indicated when the number of germ cells between groups differs significantly (p<0.05). NS: not statistically significant. Unpaired Student's *t*-test.

The online version of this article includes the following figure supplement(s) for figure 5:

**Figure supplement 1.** *ndrg1b* mutation did not alter the 11-ketotestosterone (11-KT) levels in medaka adults.

**Figure supplement 2.** *ndrg1b* is expressed only in presumptive A-type spermatogonia (SG) from the testis.

cystic proliferation, with the concomitant lower sperm count, have direct influence on reproduction and mating behavior.

Although we observed that the mutated individuals did not present changes in secondary sexual characteristics, such as the separation of hindmost rays from other rays in the dorsal fin (*Figure 4E–H*), which is strongly related to the presence of androgens in males (*Ngamniyom et al., 2009*), the corroboration of androgen levels is necessary because they are related to reproductive behavioral changes in adults of medaka (*Hiraki-Kajiyama et al., 2019*). The 11-ketotestosterone (11-KT) quantification did not show differences between wt and *sgN1b* males in a reproductive trial (*Figure 5—figure supplement 1A, B*), supporting that reduction in cystic proliferation and a reduced number of germ cells in males would not affect sex steroid-producing somatic cells.

Considering the unchanged levels of androgens in the *ndrg1b* mutants but significant changes in reproductive behavior, we proceeded to analyze whether this gene is expressed in other extragonadal tissues, specifically in the brain. We observed that the expression of *ndrg1b* is restricted to the gonad (*Figure 5—figure supplement 2A, B*), not being observed in the brain (*Figure 5—figure supplement 2A, E*). Within the testis, *ndrg1b* was observed in presumptive A-type undifferentiated spermatogonia (SGa; *Figure 5—figure supplement 2B*) since the cells that presented *ndrg1b* marking are situated in the most peripheral region of the tubule, where *oct4*-positive cells (*Figure 5—figure supplement 2C*) or SGa are observed (*Figure 5—figure supplement 2D*).

## Discussion

*Ndrg1* homologs have been found in different species of invertebrates and vertebrates (*Fang et al., 2014*; *Kovacevic et al., 2013*; *Sun et al., 2013*). In mammals, it has been well established that *Ndrg1* exerts an inhibitory effect on cancer cell proliferation (*Chang et al., 2014*; *McCaig et al., 2011*; *Xi et al., 2017*; *Zhang et al., 2019*). Additionally, several studies have shown that *Ndrg1* is expressed in the gonads of humans, mice, sheep, and medaka (*Burl et al., 2018*; *Kikuchi et al., 2019*; *Lachat et al., 2002*; *Qu et al., 2019*; *Stévant and Nef, 2019*); however, its role in gonadal function and reproduction is still unknown. In medaka, there are two paralogs of *ndrg1*: *ndrg1a* and *ndrg1b*, although only *ndrg1b* is expressed in the gonad. Interestingly, *ndrg1b* expression is opposite to EGSC cystic proliferation in both sexes. Present from invertebrates to vertebrates, cystic division is a highly conserved mechanism that precedes gametogenesis and is crucial to reproductive success. However, its molecular regulation remains unknown (*Dinardo et al., 2011*; *Hinnant et al., 2017*; *Narbonne-Reveau et al., 2006*).

To study the participation of *ndrg1b* in cystic proliferation, our first idea was to generate the loss of function of this gene through the generation of a homozygous mutant. However, obtaining a mutant line for *ndrg1b* was not possible because in homozygosis it would appear to be lethal. From the analysis of F2 individuals, and waiting for the presence of 25% homozygous KOs to *ndrg1b*, surprisingly, this genotype was not obtained because individuals died during embryonic early development for unknown reason. Moreover, it is important to highlight that obtaining these F1 and F2 required a significant investment of time given the very low spawning rate of the heterozygote females, the low fertility of heterozygote males, and the low hatching rate. The impossibility of obtaining a stable line for the loss of function of *ndrg1b* led us to search for another robust and reproducible alternative.

In fish, and especially in medaka (*Ansai and Kinoshita, 2014*), it has been well established that the complex Cas9/sgRNA generated a high-efficiency biallelic mutations of injected animals (cas9 +sgRNA). The high efficiency of biallelic mutant and the use of this approach to research in medaka were already corroborated recently in our laboratory (*Castañeda Cortés et al., 2019*). This high efficiency is observed in the present work too, in which two different sgRNAs used to generate the biallelic mutant of *ndrg1b* presented 96.6% for sg1_ndrg1b and 80% for sg2_ndrg1b. Despite this high efficiency, this could not translate into a robust and reproducible phenotype, so were performed different controls. In this regard, we analyzed the gonadal phenotype of a second sgRNA (sg2-ndrg1b) injection, obtaining the same reduction of cystic proliferation of germ cells and adding to the verification of the absence of off-target, it supports the high directionality of the CRISPR/Cas9 system to generate the loss of function through the generation of biallelic mutants for *ndrg1b*. Another important concern to analyze is the reproducibility of the chosen system. On this regard, the phenotype throughout the different experiments, both in embryonic, juveniles, and in adult stages, carried out

in many different individuals, supports the high repeatability and robustness of the phenotype of the biallelic mutant of *ndrg1b*. Furthermore, in a previous study by *Castañeda Cortés et al., 2019* in medaka using biallelic mutants of two different sgRNAs targeting other loci, the corticotropin-releasing hormone receptors (crhr1 and crhr2), no relevant gonadal phenotype was observed in biallelic mutant fish reared at 24°C, showing the lack of a relevant phenotype in the germline after the use of CRISPR/Cas9. In addition, and something that is a strong point of the present work, this phenotype of the biallelic mutant was also analyzed and observed in both sexes, which even present different developing times in cystic proliferation.

In the present study, we generated *ndrg1b* F0 mutants by using CRISPR/Cas9 technology and showed that the lack of *ndrg1b* leads to a reduction in the total number of germ cells by inhibiting cystic proliferation in both sexes during early gonadal development and adulthood. This is evidence that *ndrg1b* is likely involved in the molecular regulation of cystic division in medaka. Gametogenesis commences within the spermatocysts or cysts in males and within germline cradles in females, from germline stem cell precursors called spermatogonia or oogonia, respectively (*Nishimura et al., 2016*). Histological examination of adult gonads showed that medaka *ndrg1b* mutants displayed typical female and male gonadal structure with all the different stages of germ cells; however, they had a lower production of gametes. In this regard, although a previous study has shown that type I proliferation is important to germline stem cell number in adult medaka gonads (*Morinaga et al., 2007*), here we showed that type II proliferation (cystic) is also crucial to gamete production, and consequently, to related reproductive parameters such as the number of spawned eggs, spermatozoa, and fertilization rate. Moreover, expression of *ndrg1b* in testis is restricted to the most peripheral region of the tubule, where A-type undifferentiated spermatogonia (SGa) are present (*Iwai et al., 2006*). This location of *ndrg1b* expression is in agreement with the reduction of cystic proliferation since it is not observed in B-type spermatogonia (SGb), which have already entered that proliferation.

Additionally, we observed that the reduction of cystic proliferation in *ndrg1b* mutants did not induce any sex reversal or modification in the sex ratio. These observations are in agreement with *Nishimura et al., 2018* in *figla*, *meioC*, and *dazl* mutants in medaka, in which follicle formation is disrupted, germ cells are unable to commit to gametogenesis, and germ cells do not develop into EGSCs, respectively. All these mutants exhibited female secondary characteristics and ovary structures, demonstrating that the mechanism underlying sex reversal is manifested before germline cells exit from the status of stem cells/gonocytes and enter the gametogenic program (*Nishimura et al., 2018*). In contrast, medaka mutants that had an effect on type I proliferation, like the germ cell-deficient *cxcr4* (*Kurokawa et al., 2007*) and also loss of *amhrII* (TGF-β pathway member) that display hyperproliferative germ cells (*Morinaga et al., 2007*; *Nakamura et al., 2012*), showed sex reversal. Interestingly, our results showed that the TGF-β pathway is not involved in the cystic division, and the increased proliferation in males is due to inhibition of TGF-β members involved in controlling germ cell type I proliferation, such as *amh*. Further studies are therefore necessary to elucidate the regulatory pathways of this type of proliferation.

An interesting observation in the present work was the lower spawn by normal females paired with *ndrg1b* mutant males. It has been established that sexual behavior is linked to reproductive success (*Oshima et al., 2003*), and, fortunately, the sexual behavior of medaka consists of a sequence of several actions that are easily quantified (*Ono and Uematsu, 1957*; *Walter and Hamilton, 1970*). We specifically observed differences in male mating behavior related to stimulation of female spawning and fertilization, such as the duration of quivering and number of convulsions; actions that seem to be conserved in vertebrates because ejaculation in male rats is accompanied by strong vibration and significant abdominal contraction (*Qin et al., 2017*). Sexual behavior has been found to be strongly controlled by the brain (*Hiraki-Kajiyama et al., 2019*; *Mitchell et al., 2020*; *Yang and Shah, 2014*) and levels of sex steroid hormones (*Hiraki-Kajiyama et al., 2019*; *Munakata and Kobayashi, 2010*). Here, it is important to note that in medaka the secondary sex characters are induced by sex steroid hormones (*Kikuchi et al., 2019*; *Sato et al., 2008*). *ndrg1b* mutants showed normal secondary sex characters in both males and females. Moreover, the level of the main androgen in fish, 11-KT, did not show differences between wt and *ndrg1b* mutants, thus establishing that sex steroid synthesis is not impaired in these animals. Regarding sexual behavior, in studies where the regulation of sex hormones was altered, changes in sexual behavior were primarily related to courtship, such as the frequencies of following, dancing, and latency from the first courtship (*Hiraki-*

*Kajiyama et al., 2019*; *Oshima et al., 2003*). Additionally, when brain regulators were altered, such as arginine-vasotocin (*Yokoi et al., 2015*), oxytocin ligand (oxt) (*Yokoi et al., 2020*), and TN-GnRH3 (*Okuyama et al., 2014*), changes in courtship display and sexual motivation, latency to mating and number of courtships displays, and latency to mating were observed, respectively. None of these courtship actions changed in the male *ndrg1b* mutants, leading us to consider whether the specific mating actions of quivering and convulsions are exclusively related to the lower number of spermatogonia and spermatozoa in males. To our knowledge, only one other study has observed this loss of mating vigor in medaka. In it, successive multiple mating-induced depletion of sperm reserves decreased fertilization and mating rate, and even females responded by reducing clutch size (*Weir and Grant, 2010*). In this respect, this association warrants further investigation in vertebrates including brain, endocrine, and gonadal features.

Taken together, our data show that mutations of *ndrg1b* lead to disruption of germline stem cell cystic proliferation with a subsequent reduction in the number of gametes and an unexpected change in mating behavior of medaka. Altogether, these data are significant for three main reasons: (1) we showed that *ndrg1b* is involved in the regulation of cystic proliferation during different stages of medaka development (embryo, juvenile, and adult) in a TGF-β-independent manner; (2) we demonstrated that mutation of *ndrg1b* leads to changes in number of gametes, decreasing reproductive success for both sexes; and (3) *ndrg1b* mutation affected male mating behaviors in medaka without changing androgen production.

## Materials and methods

### Source of medaka

All experiments were performed with medaka (*Oryzias latipes*) (strain hi-medaka, ID: MT835) supplied by the National BioResource Project (NBRP) Medaka (http://www.shigen.nig.ac.jp/medaka/). Fish were maintained and fed following standard protocols for medaka (*Kinoshita et al., 2012*). Fish were handled in accordance with the Universities Federation for Animal Welfare Handbook on the Care and Management of Laboratory Animals (http://www.ufaw.org.uk) and internal institutional regulations. Fertilized medaka eggs were incubated in 60 mm Petri dishes with embryo medium (17 mM NaCl, 0.4 mM KCl, 0.27 mM CaCl$_2$·2H$_2$O, and 0.66 mM MgSO4; pH 7) until hatching and subsequently in 2 l water tanks to adults under a constant photoperiod (14L:10D) in a closed circulation water system at a controlled temperature (26 ± 0.5°C).

### Sample collection

Samplings were performed at different stages of medaka development: embryonic stages 35 and 39 (5 and 9 days post fertilization, respectively) and juvenile–adult stages 10, 20, and 80 dph, according to a previous description of medaka development (*Iwamatsu, 2004*). These stages are important for medaka gonadal development. Stage 35 corresponds to a sexually undifferentiated gonad in both sexes and only type I divisions (first mitotic proliferation). This stage precedes the beginning of type II division (cystic proliferation) in XX gonads (stages 36–39). Stage 39 corresponds to the maximum EGSC proliferation in XX embryos, and involves sexually dimorphic gonads, just at the time of hatching. At 10 dph, type II division is present only in XX gonads, while in XY gonads, it begins only at 20 dph. At 80 dph, animals have become sexually mature adults. For all sample collections, fish were euthanized by immersion in tricaine at 30 ± 50 mg/l and processed according to the technique to be used. To determine genotypic sex, DNA from the tail and head of each animal was analyzed by PCR to determine the presence of the *dmy/dmrt1bY* gene; ß-actin gene was used as a DNA loading control (*Supplementary file 1A*; *Nanda et al., 2002*). The PCR products were analyzed on 1% agarose gel.

### Whole-mount RNA in situ hybridization

Whole-mount RNA in situ hybridizations were performed as previously described (*Nakamura et al., 2006*). Fluorescein isothiocyanate (FITC)- and digoxigenin (DIG)-labeled probes were synthesized from the full-length medaka cDNA of *ndrg1a*, *ndrg1b, and oct4* (stem cell marker also known as *pou5f1*; *Wang et al., 2011*) using pGEM-T Vector (Promega) linearized plasmid. Embryos at stage 35, and male brains and testis were fixed overnight in 4% RNAse-free paraformaldehyde at 4°C,

permeabilized using 20 µg/µl proteinase K at room temperature (RT), and hybridized at 68°C overnight with either *ndrg1a*, *ndrg1b*, or *oct4* DIG-labeled RNA probes, or at 66°C overnight with *ndrg1b* FITC-labeled RNA probe. Hybridized DIG probes were detected using an alkaline phosphatase–conjugated anti-digoxigenin antibody (1:2000; Roche) in the presence of nitro blue tetrazolium/5-bromo-4-chloro-3′-indolyphosphate substrates (Roche). Stained embryos were embedded in gelatin, cryostat sectioned at 14–16 µm thickness, and photographed. Hybridized FITC probes were examined under a fluorescence microscope.

## RNA quantification by RT-qPCR

Embryos at stages 35 and 39, juveniles at 10 and 20 dph, and tissue of male brains and testis were used for gene expression analysis. For this purpose, the gonadal region of the body was immediately frozen in liquid nitrogen and subsequently stored at −80°C until RNA extraction. Once the sex of each individual was known by PCR, total RNA was extracted from a pool of five embryos, or individually from juveniles, using 300 µl of TRIzol Reagent (Life Technologies) following the manufacturer's instructions. RNA from each sample (500 ng) was used to perform cDNA synthesis, following previous studies (*Castañeda Cortés et al., 2019*). Real-time PCR primers are listed in *Supplementary file 1A*. Gene-specific RT-PCR was performed using the SYBR green master mix (Applied Biosystem) and Agilent MX3005P Multiplex QPCR Real-time Thermal Cycler (Stratagene). The amplification protocol consisted of an initial cycle of 1 min at 95°C, followed by 10 s at 95°C and 30 s at 60°C for a total of 45 cycles. The subsequent quantification method was performed using the 2-ΔΔCt method (threshold cycle; assets; thermofisher.com/TFS-Assets/LSG/manuals/cms_040980.pdf) and normalized against reference gene values for ribosomal protein L7 (*rpl7*) (*Zhang and Hu, 2007*) or elongation factor 1 alpha (*ef1α*) (*Hirayama et al., 2006*).

## TGF-β inhibitor treatment

To analyze whether TGF-β regulates *ndrg1b* expression, A83-01 (Sigma-Aldrich, USA), a selective TGF-β inhibitor was used (*Tojo et al., 2005*). A83-01 inhibits the phosphorylation of Smad2, thereby blocking the signaling of TGF-β type I receptor (*Tojo et al., 2005*). A83-01 was first dissolved in DMSO (3 mM) and subsequently diluted in embryo medium (17 mM NaCl, 0.4 mM KCl, 0.27 mM CaCl$_2$·2H$_2$O, and 0.66 mM MgSO4; pH 7) to obtain a solution of 3 µM. Fertilized medaka eggs were incubated in 70 mm Petri dishes with or without 3 µM A83-01 from stage 35 to stage 39 at 26°C (*Figure 1—figure supplement 1*). Freshly made dilutions were used, and the medium was changed every 24 hr. As control, a group of embryos were incubated only with 0.1% DMSO. The concentration used in this study was based on *Tojo et al., 2005*. After the incubation period, embryos were collected at stage 39 for sex genotyping, RT-qPCR, and cell quantification by histological analysis, as described in this article.

## Generation of *ndrg1b* mutants using CRISPR/Cas9-induced mutagenesis

CRISPR/Cas9 target sites were identified and designed using the medaka genome reference available on the Ensembl genome database (ENSORLG00000004785) and the CCTop-CRISPR/Cas9 target online predictor (CCTop, http://crispr.cos.uni-heidelberg.de/) (*Stemmer et al., 2015*). Two sequences 5′GG-(N18)-NGG3′ in exon 6 of *ndrg1b* were identified. The selected target sites are shown in Fig. S1A and Fig. S2A. Each sgRNA was synthesized according to a protocol previously established by *Ansai and Kinoshita, 2014*. Briefly, the pair of inverse-complementary oligonucleotides was annealed and cloned into the pDR274 vector (Addgene #42250) in the BsaI cloning site. The modified pDR274 vectors were digested with DraI, and each *ndrg1b*-specific guide RNA (SgRNA_*ndrg1b*) was transcribed using the MEGAshortscript T7 Transcription Kit (Thermo Fisher Scientific). The synthesized sgRNAs were purified by RNeasy Mini kit purification (Qiagen). For the Cas9-RNA in vitro synthesis, the pCS2-nCas9n vector (Addgene #47929) was linearized with NotI, and capped Cas9 RNA was transcribed with the mMessage mMachine SP6 Kit (Thermo Fisher) following the manufacturer's instructions and purified with the RNeasy Mini kit (Qiagen). Cas9-RNA (100 ng/µl) and SgRNA_*ndrg1b* (50 ng/µl) were co-injected into one-cell stage embryos as described previously (*Kinoshita et al., 2000*). Embryos injected only with *cas9* mRNA were used as controls (wt). Microinjections were performed with a Nanoject II Auto-Nanoliter Injector (Drummond Scientific) coupled to a stereomicroscope (Olympus).

## Biallelic CRISPR mutants screening and off-target analysis

To analyze the efficiency and specificity of the CRISPR/Cas9 system, genomic DNA was extracted with alkaline lysis buffer using 3 dpf embryos, as described previously (*Ansai and Kinoshita, 2014*), DNA was used as a template for PCR prior to HMA using the primers listed in *Supplementary file 1A*. Electrophoresis was performed on 10% acrylamide gels and stained with ethidium bromide (*Ota et al., 2013*). Multiple heteroduplex bands shown by HMA in PCR amplicons from each injected embryo were quantified as embryos with biallelic mutations, whereas single bands were quantified as non-edited embryos. The mutation rate was calculated as the ratio of the number of multiple heteroduplex bands shown in PCR amplicons from each Cas9-sgRNA-injected embryo to the sum of all embryos injected multiplied by 100 (n = 20/per sgRNA) (*Ansai and Kinoshita, 2014*). Sg1RNA_*ndrg1b* was selected for subsequent phenotypic analysis due to its greater efficiency. Additionally, potential off-target sites in the medaka genome for Sg1RNA_*ndrg1b* were searched for by using the CCTop-CRISPR/Cas9 target online predictor (*Stemmer et al., 2015*). All potential off-target sites identified were analyzed by HMA using the primers listed in *Supplementary file 1A*. The screening of indels was performed in F1 fish. Biallelic mutant adult (F0) medaka were mated with wt medaka. Genomic DNA was extracted from each F1 embryo for analysis of mutations by HMA as described previously (*Supplementary file 1A*). Mutant alleles in each embryo were determined by sequencing of the *ndrg1b* genomic DNA region. All phenotypic analyses were performed using positive *ndrg1b* biallelic mutants F0 injected with Cas9 and Sg1RNA_*ndrg1b* (*SgN1b*) and Cas9-injected individuals as controls (wt). Additionally, with the goal of analyzing KO individuals, an F2 generation was obtained by mating F1 fish harboring the same mutation (the Δ31, deletion of 31 bp) with each other; the obtained F2 individuals were analyzed by HMA genotyping using genomic DNA extracted from each fin clip of F2 80 dph fish as a template for PCR prior to first HMA using the primers listed in *Supplementary file 1A*, following the protocol previously established by Ansai and Kinoshita (*Hatada, 2017*). For reliable distinction between wt and KO fish (both homozygous show a single band pattern in first HMA), each PCR product from the first HMA was mixed with a wt template separately, then reannealed by heating at 95°C for 5 min and gently cooled to RT to perform the second HMA, where the wt will show only one band, while KO fish will show a multiple banding pattern similar to heterozygotes (*Hatada, 2017*; *Figure 2—figure supplement 3*).

## Immunofluorescence analysis

Gonadal regions from individuals at stage 39 were processed, fixed in Bouin's solution, embedded in paraffin, and sagittal sectioned at 5 µm. Sections were washed with 0.1 M phosphate-buffered saline (PBS pH 7.4) and blocked in 0.1 M PBS containing 0.5% bovine serum albumin (Sigma-Aldrich) and 0.5% Triton X-100 for 60 min before overnight incubation with primary antibody anti-OLVAS (1:200, rabbit, Abcam 209710) and anti-PCNA (1:200, mouse, Sigma P8825) at 4°C. After incubation, the sections were washed twice in PBS and incubated at RT for 90 min with Alexa Fluor 488-conjugated goat anti-rabbit IgG (Thermo Fisher Scientific, A-11008) and 594-conjugated goat anti-mouse IgG (Thermo Fisher Scientific, A32742) secondary antibodies at a dilution of 1:2000 in PBS. After incubation, sections were rinsed twice with PBS and mounted with Fluoromount mounting medium (Sigma-Aldrich) containing 4',6-diamidino-2-phenylindole (DAPI, 5 µg/ml, Life Technologies).

## TUNEL assay

The presence of apoptosis at stage 39 was detected through the In situ Cell Death Detection Kit, Fluorescein (Roche). Samples fixed in Bouin's solution, embedded in paraffin, and sagittal sectioned at 5 µm were treated according to the manufacturer's manual, with a step of permeabilization with 0.1% Triton X-100, 0.1% sodium citrate in PBS 1× solution. After labeling the reaction for 1 hr at 37°C, sections were immunostained with OLVAS, as described above.

## Histological analysis

Gonadal samples from juveniles at 20 (from whole body) and 80 dph (from gonadal tissue) were fixed in Bouin's solution, embedded in paraffin, sectioned at 5 µm, and subjected to standard hematoxylin and eosin staining. For the TGF-β inhibitor treatment, stage 39 individuals were fixed in 4% buffered glutaraldehyde, embedded in Technovit (7100-Heraeus Kulzer), sectioned at 5 µm

thickness, and stained with 0.1% toluidine blue to quantify the germ cells using a light microscope (Leica DM6000 BD, Leica Microsystems).

## Cell quantification

All section photographs were taken using a ZEISS Axio Observer 7 Led colibri 7, Nikon Eclipse E600 and Nikon Digital Sight DSQi1Mc. Images were analyzed using FIJI software (https://imagej.nih.gov/ij/). Individual cells were counted manually with the Cell Counter plugin for FIJI. Type I and II germ cells were counted as previously described (*Saito et al., 2007*), where single isolated germ cells were counted as type I, while clusters with more than two germ cells were counted as germ cells undergoing type II division. Spermatogonia were counted based on the description by *Iwasaki et al., 2009* and oocytes based on *Seki et al., 2004* and *Gay et al., 2018*. All gonadal sections were counted for stage 39 individuals. For 20 and 80 dph individuals, three gonadal sections were counted, setting the distance between each section to avoid counting the same cell twice. Each section was counted two times in all experiments to reduce technical errors.

## Sperm quantification

Sperm was collected from mutant and control adult males after immersion in tricaine at $30 \pm 50$ mg/l. Each male was placed on a foam rubber holder with the belly facing up; the urogenital pore was dried and sperm was collected with a micropipette while gentle pressure was applied from bilateral abdominal toward the anal opening using fine forceps. Prior to the analysis, the sperm was diluted 5000 times in cold Ringer's solution (NaCl 0.75%, KCl 0.02%, $CaCl_2$ 0.02%, $NaHCO_3$ 0.002%) and kept at 4°C. Sperm counts were determined by hemocytometer (Bright-Line, Hausser Scientific) analysis under light microscope at $\times 200$ magnification, performing three counting replicates per individual.

## Evaluation of reproductive success

To evaluate reproductive success, we used three categories of sexually mature couples: control (wt female and male), mutant female (wt male and *ndrg1b* mutant female), and mutant male (wt female and *ndrg1b* mutant male). Each couple was acclimated for 5 days in a breeding tank to ensure and standardize their reproductive conditions. For 10 consecutive mornings, each couple's fertilized and unfertilized eggs were carefully collected by hand, placed in a separate dish, and counted to calculate total spawned eggs, fertilization (number of fertilized eggs/number spawned eggs), and hatching (number of hatched larvae/number of fertilized eggs).

## Sexual behavior assay

Sexual behavior assay was performed over 10 days using two categories of sexually mature couples, control and mutant male, mentioned above, which were allowed to acclimate for 5 days to ensure and standardize their reproductive conditions. On the remaining 5 days, the procedure was performed as previously described (*Okuyama et al., 2014*). Males and females were separated in the evening (6–7 PM) the day before the assay using a transparent plastic cup with small holes for water exchange. The following morning the mating pairs were placed together in a single transparent tank and their sexual behavior was recorded for 20 min using a digital video camera (*Figure 5A*). The recording order was randomly changed every day. Water inflow to the tanks was stopped during recording. Females that did not spawn or spawned without any mating behavior were excluded from the analysis. The following behavioral parameters were calculated from the video recordings: latency until first courtship display, latency from the first courtship display to the wrapping that resulted in spawning; number of wrappings before and after spawning, the duration of wrapping, and within wrapping time; and the duration of quivering and number of convulsions. Each action during sexual behavior was identified following the descriptions by *Ono and Uematsu, 1957* and *Walter and Hamilton, 1970* and analyzed twice by two different authors of the present study who did not know the category of the couple.

## Blood levels of 11-KT in sexually mature males

Plasma 11-KT concentrations were measured using an enzyme-linked immunosorbent assay kit (Cayman 582751) according to the manufacturer's instructions. Briefly, control and *sgN1b* sexually

mature males (n = 7) were coupling with wt females (*Figure 5—figure supplement 1A*). Males and females were separated in the evening (6–7 PM) the day before the assay using a transparent plastic cup with small holes for water exchange, and then blood samples of anesthetized male individuals were collected directly from the caudal vein using a glass capillary tube (*Royan et al., 2020*). All fish were recovered after blood extraction. Steroid extractions were performed twice with ethyl acetate/ hexane from 1 µl of whole blood and then reconstituted in 200 µl of assay buffer. All samples were assayed in duplicate, and hormone levels were determined based on a standard curve.

## Statistical analysis

Values are presented as mean ± standard error of the mean for continuous variables and as percentages for categorical variables. Fold change and statistical analysis of RT-qPCR quantifications were performed using FgStatistics software (http://sites.google.com/site/fgStatistics/), based on the comparative gene expressions method (*Pfaffl, 2001*). Statistical analyses were performed by using GraphPad Prism (GraphPad Software, San Diego, CA). Continuous variables were compared between two groups by the unpaired two-tailed Student's *t*-test. If the *F*-test indicated that the variance differed significantly between groups, Welch's correction to the Student's *t*-test was employed. For more than two groups, continuous variables were compared by one-way analysis of variance, followed by Tukey's multiple comparisons test, comparing the mean of each column with the mean of every other column. All differences were considered statistically significant when $p < 0.05$.

## Acknowledgements

We thank Tech. Gabriela C López and Dr. Leandro A Miranda (INTECH) for helping with histological preparations. We also thank Dr. Tania Rodriguez for technical support and Tech. Javier Herdman (INTECH) for fish handling. We are grateful to NBRP Medaka (https://shigen.nig.ac.jp/medaka/) for providing HNI (Strain ID: MT835). We thank Dr. John M Murray (Biodesign Center for Mechanisms of Evolution, Arizona State University) for critical reading of the manuscript. This work was supported by the Agencia Nacional de Promoción Científica y Tecnológica Grant 0366/12 and 2501/15 (to JIF). RHN and IFR were supported by São Paulo Research Foundation (FAPESP), Brazil (Grants 14/07620-7 and 18/10265-5), CONICET and Sao Paulo Research Foundation (FAPESP), and International Cooperation Grant D 2979/16 (to JIF and RSH). LFAP, DCCC, and ODMA were supported by a PhD scholarship from the National Research Council (CONICET). JIF is a member of the Research Scientist Career at the CONICET.

## Additional information

### Funding

| Funder | Grant reference number | Author |
| --- | --- | --- |
| Agencia Nacional de Promoción Científica y Tecnológica | 0366/12 | Juan I Fernandino |
| Agencia Nacional de Promoción Científica y Tecnológica | 2501/15 | Luisa F Arias Padilla |
| Fundação de Amparo à Pesquisa do Estado de São Paulo | 14/07620-7 | Ivana F Rosa Rafael H Nóbrega |
| Fundação de Amparo à Pesquisa do Estado de São Paulo | 18/10265-5 | Ivana F Rosa Rafael H Nóbrega |
| Consejo Nacional de Investigaciones Científicas y Técnicas | CONICET and São Paulo Research Foundation International Cooperation Grant D 2979/16 | Ricardo S Hattori Juan I Fernandino |
| São Paulo Research Foundation | CONICET and São Paulo Research Foundation International Cooperation Grant D 2979/16 | Ricardo S Hattori Juan I Fernandino |

The funders had no role in study design, data collection and interpretation, or the decision to submit the work for publication.

## Author contributions
Luisa F Arias Padilla, Conceptualization, Data curation, Formal analysis, Methodology, Writing - review and editing; Diana C Castañeda-Cortés, Data curation, Formal analysis, Methodology; Ivana F Rosa, Data curation, Methodology; Omar D Moreno Acosta, Ricardo S Hattori, Methodology; Rafael H Nóbrega, Investigation, Methodology, Writing - review and editing; Juan I Fernandino, Conceptualization, Formal analysis, Supervision, Funding acquisition, Investigation, Methodology, Writing - original draft, Project administration, Writing - review and editing

## Author ORCIDs
Luisa F Arias Padilla (iD) https://orcid.org/0000-0003-4689-2561
Juan I Fernandino (iD) https://orcid.org/0000-0003-1754-2802

## Decision letter and Author response
Decision letter https://doi.org/10.7554/eLife.62757.sa1
Author response https://doi.org/10.7554/eLife.62757.sa2

# Additional files

## Supplementary files
• Supplementary file 1. (**A**) Primers sequences, ENSEMBL accession numbers and respective references of each gene were added. (**B**) Sex ratio of both sexes embryos injected with cas9 (wildtype) and the sgNb1 (cas9+sg1_ndrg1b).
• Transparent reporting form

## Data availability
All data generated or analysed during this study are included in the manuscript and supporting files.

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
