## [Decision Letter]

**Acceptance summary:**

In the manuscript by Arias Padilla et al. the authors investigate the role of ndrg1b in gonad development and mating behavior in medaka. They find ndrg1b is required for cystic proliferation of the gonad and for gamete production, reproductive success and mating vigor. This study generates several insights into gonad development and links defects in gamete production to mating behavior and reproductive success in a way that is not obviously rooted in hormone production.

**Decision letter after peer review:**

Thank you for submitting your article "Cystic proliferation of germ stem cells is necessary to reproductive success and normal mating behavior in medaka" for consideration by *eLife*. Your article has been reviewed by three peer reviewers, including Michel Bagnat as the Reviewing Editor and Reviewer #1, and the evaluation has been overseen by Marianne Bronner as the Senior Editor.

The reviewers have discussed the reviews with one another and the Reviewing Editor has drafted this decision to help you prepare a revised submission.

Summary:

In the manuscript by Arias Padilla et al. the authors investigate the role of ndrg1b in gonad development and mating behavior in medaka. They find ndrg1b is required for cystic proliferation of the gonad and for gamete production, reproductive success and mating vigor.

This is an interesting and straightforward study that reports a novel function for ndrg1 in gonad development and also generates several insights into gonad development and links defects in gamete production to mating behavior and reproductive success in a way that is not obviously rooted in hormone production. However, there are significant problems related to the use of F0 CRISPR injected animals that cannot be overcome unless stable mutants are analyzed. There is also a need to assess hormone production and provide additional evidence that might help exclude extra-gonadal function for ndrg1b that could explain the behavioral phenotypes.

We realize revisions would not be feasible within a reasonable time unless establishment of a stable mutant line has already been in progress. The authors should evaluate whether under current conditions the essential revisions listed below are feasible.

Essential revisions:

1) Reanalyzing the phenotypes using stable mutant fish. Both male and female mutants appear to be fertile, considering that they can produce fertilizable gametes and their offspring can develop normally. Even though male mutants showed lower fertilization and hatching rates, it would thus be possible to generate a stable mutant line and assess the phenotypes of the siblings from the line.

2) Measuring hormonal production. If the androgen levels are altered in the mutants, the conclusion that the reduced number of germ cells leads to the behavioral deficit would be wrong. Thus, the circulating levels of sex steroids should be directly assessed.

3) Confirming that ndrg1b functions exclusively in the gonad. While the ideal experiment would be to demonstrate that gonad specific rescue restores mating performance, a careful gene expression analysis using in situ hybridization would help support the claims.

4) Recommendations for behavioral assays. The medaka strain used in this study (himedaka) is not inbred and exhibits large interindividual genetic variations. In addition, the number of germ cells varies depending on environmental conditions early in life. Therefore, phenotypes should be compared between homozygous knockouts and wild-type/heterozygous siblings that were derived from the same clutch of eggs and reared in the identical environmental condition (after establishing a knockout medaka line) to control for genetic background and environmental effects.

[Editors' note: further revisions were suggested prior to acceptance, as described below.]

Thank you for resubmitting your work entitled "Cystic proliferation of germ stem cells is necessary to reproductive success and normal mating behavior in medaka" for further consideration by *eLife*. Your revised article has been evaluated by Marianne Bronner (Senior Editor) and a Reviewing Editor.

Summary:

Reviewers agree in that he manuscript has been improved, but there are some remaining issues that need to be addressed before acceptance. Specifically, controls are need for the F0 CRISPRs and the reported lethality numbers need to be re-examined and/or explained. Additionally, please include a discussion point justifying the use of F0 CRISPRs mentioning the use of a second gRNA and summarizing controls. Along with two experimental points that need to be addressed, there are also several suggestions for editorial changes, editing by a professional or a native speaker would help improve the manuscript and make the work easier to read.

1) While their results suggest that the use of F0 animals is viable in this case, particularly considering they are investigating a mutation that affects the germ line, additional controls will be needed to rule out non-specific defects due to genome editing. Controls may include: 1-documenting the absence of relevant phenotypes following the use of an inefficient or inactive gRNA targeting ndrg1b; 2-documenting the absence of relevant phenotypes following the use of a gRNA targeting another locus; 3-testing for toxicity by inject the gRNA or Cas9 alone.

2) A) If the homozygous mutants are lethal, the ratio of wild-type fish to heterozygous fish in progeny should be 1:2 when heterozygous mutants are crossed to each other. However, this was not the case, and the ratio was almost 1:1. Assuming that one parent of the crossed F1 generation fish was the wild-type, the result can be well explained. Did the authors confirm the result that the homozygous mutants cannot be obtained by crossing heterozygous mutants, for example, by conducting a follow-up test using another parent fish?

B) The methodology of the 2nd HMA, whose results are shown in Figure 2—figure supplement 3, panel D, is not described anywhere. Therefore, it is not possible to see whether the method was designed to detect the homozygous mutants correctly.

[Editors' note: further revisions were suggested prior to acceptance, as described below.]

Thank you for resubmitting your work entitled "Cystic proliferation of germline stem cells is necessary to reproductive success and normal mating behavior in medaka" for further consideration by *eLife*. Your revised article has been evaluated by Marianne Bronner (Senior Editor) and a Reviewing Editor.

The text changes improved the manuscript but there are two important remaining issues that were raised on the original manuscript and the revised version that need to be addressed. Resolving these issues is essential for ensuring that sufficient minimal quality controls have been made before the manuscript can be accepted.

The outstanding issues are:

1) Additional controls for the F0 CRISPRs were requested. Instead of performing additional injections of gRNA alone and documenting the absence of relevant phenotypes upon editing of another locus, literature was cited and it was the point was argued, but without including any discussion points in the manuscript. 1A- If the analyses in Castaneda-Cortes et al. is sufficient for claiming that genome editing of another locus does not lead to the type of germ line defects reported in the present manuscript, a discussion point citing this reference should be added along with a mention of the use of another gRNA targeting ndrg1b and reproducing the reported phenotype, the lack of phenotype following injection of Cas9 mRNA alone and, if possible, the gRNA1 alone (general lack of toxicity for gRNAs cannot be assumed).

2) The authors did not document in their revised manuscript when the lethality of the homozygous mutation arises and used HMA for genotyping, which is not optimal. At the very least, they need to determine whether homozygous embryos can be generated from heterozygous crosses and attempt to approximately define the developmental window at which homozygous mutants are lost. This issue cannot be dismissed or left open for further studies because if no homozygous mutant embryos can be generated, then heterozygous germ line defects or technical problems may be present. It is recommended that a more robust method than HMA is used for genotyping the offspring of heterozygous crosses, a 31 bp deletion should be easy to detect by PCR across the locus followed by high % agarose or acrylamide gel electrophoresis.

---

## [Author Response]

Essential revisions:1) Reanalyzing the phenotypes using stable mutant fish. Both male and female mutants appear to be fertile, considering that they can produce fertilizable gametes and their offspring can develop normally. Even though male mutants showed lower fertilization and hatching rates, it would thus be possible to generate a stable mutant line and assess the phenotypes of the siblings from the line.

We understand the concern of the reviewers to suggest reanalyze of the data using a stable mutant line. In agreement with the reviewers, this strategy was the first we chose to study the participation of *ndrg1b* in the proliferation of germ cells. However, obtaining a mutant line for this gene was not possible because in homozygosis it would appear to be lethal. This depends on our previous experiments carried out, and now added such as Figure 2—figure supplement 3, where we were able to make three important observations that support the choice of the use of F0 injected fish (as known as crispant too) in the present study. In the new Figure 2—figure supplement 3, the experimental design can be observed, where the deletion ∆31 (deletion of 31 bp), previously characterized in Figure 2—figure supplement 1C, was chosen to cross F1 and obtain the loss of function of the *ndrg1b* gene. It is important to highlight that obtaining these F1 and F2 required a significant investment of time given the low spawning rate of the heterozygote females (Figure 4—figure supplement 1A), the low fertility of heterozygote males (Figure 4—figure supplement 1B) and the low hatching rate (Figure 4—figure supplement 1C). From the analysis of F2 individuals, and waiting for the presence of 25% homozygous knockouts to *ndrg1b*, surprisingly, this genotype was not obtained (as can be seen in the table of Figure 4—figure supplement 1 performed by means of an HMA test of supplement figure 1C, D). This would establish that *ndrg1b* has another function in early stages of development that should be studied in future studies, since they are outside the scope of the present work.

In fish, and especially in medaka (Ansai and Kinoshita, 2014), it has been well established that the complex Cas9/sgRNA generated a high efficiency biallelic mutations of injected animals (cas9+sgRNA). The high efficiency of biallelic mutant and the used of this approach to research in medaka was already corroborate recently in our laboratory (Castañeda-Cortés et al., 2019). This highly efficiency is observed in the present work too, in which two different sgRNAs used to generate the biallelic mutant of *ndrg1b* presented 96.6% for sg1-ndrg1b and 80% for sg2-ndrg1b (Figure 2—figure supplement 1C, D, Figure 2—figure supplement 2B). Despite this high efficiency, each individual was checked for biallelic *ndrg1b* mutations before being used for the different analyzes for corroboration. Nevertheless, this high efficiency might not translate into a robust and reproducible phenotype, which is perhaps the question established by the reviewers; however, our data supports the use of this approach to the present study. I am going to focus on some results of this work that support this choice:

i) A good control supporting the use of F0 injected fish was analyzed the phenotype of a second sgRNA (sg2-ndrg1b). On this concern, testing different sgRNAs we obtained the same reduction of cystic proliferation of gem cells (see Figure 2B, C and Figure 2—figure supplement 2C, D), and adding to the verification of the absence of off-target, it supports the high directionality of the CRISPR/Cas9 system to generate the loss of function through the generation of biallelic mutants for *ndrg1b*. Consequently, the used of two different sgRNA presenting the same phenotype support the robustness of the chosen system.

ii) Another important point to analyze is the reproducibility of the chosen system. On this regard, the phenotype throughout the different experiments, both in embryonic stages (quantification of the number of germ cells, proliferation, apoptosis; Figure 2, Figure 2—figure supplement 2E, 4C), in juveniles (quantification of the proliferation of germ cells; Figure 3) and in adults (quantification of germ cells, quantification of parameters and reproductive tests and measure by qPCR and ELISA of ndrg1b transcripts and 11-KT respectively; Figure 4, 5 and Figure 5—figure supplement 1), carried out in many different individuals, support the high repeatability and robustness of the phenotype of the biallelic mutant of ndrg1b. In addition, and something that is a strong point of the present work, is that this phenotype of the biallelic mutant was also analyzed and observed in both sexes, which even present different developing times in cystic proliferation.

iii) As already observed in medaka (Ansai and Kinoshita, 2014; Castañeda-Cortés, et al., 2019) and confirmed by this study, the overwhelming majority of fish injected with different sgRNAs displayed normal morphological development and growth, compared with non-injected or cas9 injected embryos (Figure 4E-H). Moreover, the expression of *ndrg1b* is restricted to the gonad, at least during the gonadal development in embryo and adults (Figure 1, Figure 5—figure supplement 2). Taking together, our results support that the biallelic mutation of ndrg1b affects preferably, and perhaps exclusively, germ cells and their cystic proliferation, since type I proliferation, characteristic of asynchronous self-renewal of germ cells, is not affected (Figure 2C, 2G, 4J and 4N).

iv) Another observation supporting the use of biallelic mutant individuals is the similar phenotype observed with F1 reproductive trial (Figure 4, Figure 4—figure supplement 1). Firstly, from the beginning we had to rule out the use of F1 to carry out all the experiments of the present work, since as we explained previously, the low fertility and survival in F1 would require that a sufficient number of individuals would take an enormous amount of time, making very It is difficult to compare the data obtained in each experiment, since we would use individuals obtained sporadically over a period of time of months.

v) Other interesting observation was that the number of spawning eggs was reduced in male and female sgNb1 and F1, the fertilization rate was reduced in sgNb1 and F1 males and, interestingly, the hatching rate was sex dimorphic, and only in embryo from sgNb1 and F1 males was reduced comparing with the respective control male (Figure 4B-D, Figure 4—figure supplement 1A-C), establishing the need for future studies in this direction. Again, this robust phenotype supports the use of injected medaka individuals, at least for this gene.

In addition to the robustness and reproducibility of the data generated in the present work, the use of individuals with biallelic mutations generated by the CRISPR/Cas9 technique has generated an interesting option for the classical use of knockout in some experimental approaches. For example, a resent work in Developmental Cell of Hoshijima et al. (2019) analyzed the generation of F0 embryos to generate the loss of function of different targets using sgRNA and two-RNA component (crRNA:tracrRNA) version of the CRISPR system. They obtained similar full bi-allelic mutants using both CRISPR system, depending of the locus target. Similar results were obtained by Kroll and co-authors (2021) in a very recent paper published in *eLife*, in which standardized a simple and effective F0 knockout method for rapid screening of behavior and other complex phenotypes, obtained the same phenotype to zebrafish mutant lines. Interestingly, although they showed that with 3 gRNA they obtain a complete loss-of-function, in many genes only one was sufficient, establishing that for each gene a standardization of the phenotypes obtained is necessary, something that we efficiently do in our work. In our experiments, we chose sgRNAs that presented high efficiency of DSB, showing by HMA (Figure 2—figure supplement 1, 2). These observations are highly supported for many works in fish (e.g., Hockman et al., 2019). Moreover, the use of biallelic mutation (mosaics) has spread widely in other vertebrate models, although using different approaches but still analyzing a similar result, such as *Drosophila*, *Xenopus*, chicken, and others (Feehan et al., 2017; Naert et a., 2016; Véron et al., 2015; Williams et al., 2018), some of them published in *eLife* (Chai et al., 2020; Delventhal et al., 2019; Gandhi et al., 2020; Port et al., 2020; Sanor et al., 2020), so CRISPR / Cas has quickly become the gold standard for many experiments, outperforming pre-existing technologies such as shRNA, RNAi and Morpholinos (Evers et al., 2016; Morgens et al., 2016).

2) Measuring hormonal production. If the androgen levels are altered in the mutants, the conclusion that the reduced number of germ cells leads to the behavioral deficit would be wrong. Thus, the circulating levels of sex steroids should be directly assessed.

Initially, we observed that the mutated individuals did not present changes in secondary sexual characteristics, such as the hindmost rays in the dorsal fin, highly related to the presence of androgens in males (Ngamniyom et a., 2009), and for that reason initially we did not take into account their measurement. However, the observation made by the reviewers is valid, and the quantification of circulating androgens is needed to validate our hypothesis that changes in germ cell proliferation directly affect reproductive behavior.

To carry out the quantification of androgens, specifically 11-ketotestosterone (11-KT), the main androgen in fish (Gonçalves and Oliveira, 2011), pairs of wild males and *ndrg1b* mutants (sgNb1) with wild females were generated, which were then acclimatized for 5 days. After this and anesthetizing the fish, we proceeded to extract blood (1 μl) from the caudal peduncle vein using glass capillaries, following the protocol established by Royan et al. (2020). All fish were recovered after blood extraction. Finally, we used whole blood at a dilution of 200 μl PBS, extracting the steroid with diethyl ether and we used Cayman kit for measuring the 11-KT (see details in Materials and methods section and Figure 5—figure supplement 1A).

Finally, the quantification of 11-KT did not show differences between wild-type and sgNb1 males (Figure 5—figure supplement 1B), supporting our hypothesis that changes in mating behavior is related to a reduction in cystic proliferation and a reduced number germ cells in males.

3) Confirming that ndrg1b functions exclusively in the gonad. While the ideal experiment would be to demonstrate that gonad specific rescue restores mating performance, a careful gene expression analysis using in situ hybridization would help support the claims.

As reviewers recommended was performed extra in situ hybridization in testis and brain of male adults, observing a clear expression in testis and absent of the respective expression in brain (Figure 5—figure supplement 2B, E). Since the in situ hybridization is not a very sensitive technique for detecting small amounts of transcript, we decided to add a more sensitive technique, such as qPCR, to the reviewers' suggestion. In concordance, the presence of *ndrg1b* also was observed by qPCR only in male adult gonads (Figure 5—figure supplement 2), confirming with two different techniques the expression of *ndrg1b* exclusively in gonads.

4) Recommendations for behavioral assays. The medaka strain used in this study (himedaka) is not inbred and exhibits large interindividual genetic variations. In addition, the number of germ cells varies depending on environmental conditions early in life. Therefore, phenotypes should be compared between homozygous knockouts and wild-type/heterozygous siblings that were derived from the same clutch of eggs and reared in the identical environmental condition (after establishing a knockout medaka line) to control for genetic background and environmental effects.

In agreement with reviewers all experiments and specially the behavioral and reproductive success assays were carefully carried out under identical conditions. First of all, the individuals derived of a mating set (n=10-12) were randomly selected to the injection the same day. Subsequently injected individuals were reared at identical environmental conditions, such as light, fed, temperature and density. Additionally, fish were randomly selected to establish mating couples and the recording video order and analysis were performed as double-blind.

[Editors' note: further revisions were suggested prior to acceptance, as described below.]

Reviewers agree in that he manuscript has been improved, but there are some remaining issues that need to be addressed before acceptance. Specifically, controls are need for the F0 CRISPRs and the reported lethality numbers need to be re-examined and/or explained. Additionally, please include a discussion point justifying the use of F0 CRISPRs mentioning the use of a second gRNA and summarizing controls. Along with two experimental points that need to be addressed, there are also several suggestions for editorial changes, editing by a professional or a native speaker would help improve the manuscript and make the work easier to read.

We appreciate all previous recommendations from the reviewers, being a very good feed-back that has greatly improved the manuscript. In this present version we try to answer all the new recommendations, which can be found below.

1) While their results suggest that the use of F0 animals is viable in this case, particularly considering they are investigating a mutation that affects the germ line, additional controls will be needed to rule out non-specific defects due to genome editing. Controls may include: 1-documenting the absence of relevant phenotypes following the use of an inefficient or inactive gRNA targeting ndrg1b; 2-documenting the absence of relevant phenotypes following the use of a gRNA targeting another locus; 3-testing for toxicity by inject the gRNA or Cas9 alone.

The reviewers present a good point here; however, at point 1, we are not sure that the use of an inefficient or inactive sgRNA targeting *ndrg1b* is a good control in this case, and we will explain why. The addition of a similar molecule that does not present activity, which is an excellent control, for example, to knock-down through the use of morpholinos and for which a control morpholino is used; however, we believe that it is being carried out in our experiments by injecting Cas9 mRNA alone as control. Moreover, we also rely on the substantial evidence of the low toxicity presented by injections in stages of a cell in medaka, not being reported in the concentrations used in the present work (see reference from Ansai and Kinoshita, 2014; Fang et al., 2018; Gay et al., 2018; Homma et al., 2017; Kayo et al., 2019; Sawamura et al., 2017; Seleit et al., 2020; Yeh et al., 2017). That is why in all our experiments we performed Cas9 (RNA control) as a control, and also the manipulation control (since all the embryos were injected). Moreover, there are two other pieces of evidence that support the absence of an inactive sgRNA control. First, the use of a second sgRNA, with a different target sequence from the *ndrg1b* locus in which the same phenotype is observed (sg2-ndrg1b; Figure 2—figure supplement 2). In addition, a second evidence supporting the absent of an inactive sgRNA control can also be complemented with the response to the point 2.

The point 2, where the reviewers suggest documenting the lack of a relevant phenotype after the use of another sgRNA targeting another locus, in a previous work of our laboratory (Castañeda-Cortés et al., 2019) was used two different sgRNAs to generate biallelic mutants of corticotropin-releasing hormone receptors (crhr1 and crhr2), following the morphological analysis of the gonad at 20 dph (from this study we just take that 20 dph is the best time to analyze gonad development). From our published data, analyzing 45 individuals from two sgRNAs of crhr1 and crhr2, and both injected together, we did not observe any relevant gonadal phenotype in the biallelic mutants fish reared at 24ºC.

Regarding point 3, fortunately, the toxicity of Cas9 and sgRNA alone had been tested by Ansai and Kinoshita (2014) to different concentration and targets to the concentration used in our work. For this reason, we draw on this and other works cited above, as well as our experience. In this way, the concentrations used in the present work have been standardized, so that if, for example, a synthesized sgRNA does not generate enough indels, in general above 80%, the sgRNA is discarded, never opting for the possibility of increasing the concentration of sgRNA to obtain a higher percentage of indels, for which a greater analysis of the toxicity of these concentrations would be necessary.

2) A) If the homozygous mutants are lethal, the ratio of wild-type fish to heterozygous fish in progeny should be 1:2 when heterozygous mutants are crossed to each other. However, this was not the case, and the ratio was almost 1:1. Assuming that one parent of the crossed F1 generation fish was the wild-type, the result can be well explained. Did the authors confirm the result that the homozygous mutants cannot be obtained by crossing heterozygous mutants, for example, by conducting a follow-up test using another parent fish?

The reviewers did an excellent observation about proportion. First, all the individuals used in this experiment were controlled by a heteroduplex mobility assay (HMA) and then confirmed by sequencing. In addition, we used several pairs (n = 4), which avoid the problem of some error in their establishment. To make this confirmation, the individuals were individualized in small fish tanks until the results were analyzed, which takes around 10 days. Then, the individuals were acclimatized for a month before making the crosses.

Discarding this, is true that the Figure 2—figure supplement 3 panel B would have led to a deceive interpretation, because the percentages of heterozygous and homozygous individuals are referred to the total number of analyzed F2 individuals by HMA, and don’t to the total number of F2 progeny, that is, it was not referred to the number of fertilized eggs, and therefore the number of dead individuals is not included. This could establish that in heterozygosity the mutation is partially lethal, and more studies would be needed to confirm this.

The subtitle of Figure 2—figure supplement 3, panel B was changed: “Table indicates the obtained proportion of heterozygous and homozygous of analyzed F2 individuals by heteroduplex mobility assay.”

B) The methodology of the 2nd HMA, whose results are shown in Figure 2—figure supplement 3, panel D, is not described anywhere. Therefore, it is not possible to see whether the method was designed to detect the homozygous mutants correctly.

The reviewers are correct because the methodology didn’t well explain in Figure 2—figure supplement 3. The methodology of F2 screening was detailed in Biallelic CRISPR mutants screening and Off-target analysis – Materials and methods section: “Additionally, F2 generation was obtained by mating F1 fish harboring the same mutation (the ∆31, deletion of 31 bp) with each other, the obtained F2 trial was analyzed by HMA genotyping using genomic DNA extracted from each fin clip of F2 fish (1^st^ HMA), and to clearly discriminate homozygous mutants from wild-type fish, each PCR product from the 1st HMA was mixed with a wild-type PCR product separately, and the mixture was reannealed by heating at 95°C for 5 min and gently cooling to room temperature (2^nd^ HMA), following the protocol previously established by Ansai and Kinoshita (2017) in the book section (Hatada, 2017) (Figure 2—figure supplement 3).”

As we mentioned above the method to detect the homozygous mutant was previously stablished by Ansai and Kinoshita (2017) in the book section (Hatada, 2017) and validated by Foster (2019). Additionally, 2^nd^ HMA method has been widely used in our laboratory to the homozygous identification.

[Editors' note: further revisions were suggested prior to acceptance, as described below.]

The outstanding issues are:1) Additional controls for the F0 CRISPRs were requested. Instead of performing additional injections of gRNA alone and documenting the absence of relevant phenotypes upon editing of another locus, literature was cited and it was the point was argued, but without including any discussion points in the manuscript. 1A- If the analyses in Castaneda-Cortes et al. is sufficient for claiming that genome editing of another locus does not lead to the type of germ line defects reported in the present manuscript, a discussion point citing this reference should be added along with a mention of the use of another gRNA targeting ndrg1b and reproducing the reported phenotype, the lack of phenotype following injection of Cas9 mRNA alone and, if possible, the gRNA1 alone (general lack of toxicity for gRNAs cannot be assumed).

We appreciate the observation. It was our mistake do not add the fruitful discussion established with the reviewers in the manuscript. For this, the discussion was added to the main text (see Discussion section).

2) The authors did not document in their revised manuscript when the lethality of the homozygous mutation arises and used HMA for genotyping, which is not optimal. At the very least, they need to determine whether homozygous embryos can be generated from heterozygous crosses and attempt to approximately define the developmental window at which homozygous mutants are lost. This issue cannot be dismissed or left open for further studies because if no homozygous mutant embryos can be generated, then heterozygous germ line defects or technical problems may be present. It is recommended that a more robust method than HMA is used for genotyping the offspring of heterozygous crosses, a 31 bp deletion should be easy to detect by PCR across the locus followed by high % agarose or acrylamide gel electrophoresis.

We value the observation because we had not mentioned the timing of the F2 selection in the manuscript, and it was added to the Materials and methods section. F2 fish were analyzed by HMA at 80 dph (adult stage).

About the developmental window when the homozygotes died, we don't know the exact developmental stage, is most probable that it was around early development between 1 to 20 hours post fertilization, moment that correspond to morula to gastrula. Unfortunately, we did not explore it in more detail because is a window that generally present mortality under normal conditions in wild-type individuals. Frankly, at that time our principal objective was to obtain a high amount of homozygous individuals to be able to study cystic proliferation in fundamental stages of gonadal development. But as we added to the discussion, the generation of F1 and F2 required a significant investment of time given the very low spawning rate of the heterozygote females, the low fertility of heterozygote males and the low hatching rate, then we decided to continue with the generation of biallelic mutants. Additionally, at this current moment is not possible repeat the F2 screening to explore in detail the stage of death.

After analyze the reviewer’s recommendations we decided to delete the phrase “This suggests that *ndrg1b* has another essential function in early stages of development, which should be investigated in future studies” from the main text because the idea “*ndrg1b* has another essential function in early stages of development” is ambiguous and we don’t have enough evidences.

Looking at the observation that the reviewers make about HMA, we can infer that we have not been clear in explaining the technique. The Heteroduplex Mobility Assay method was better written in a greater detail in the Materials and methods section. This method is a practical and sensible method previously stablished for F2 screening (Hatada, 2017). In the HMA method, accurately as the reviewers mentioned in the current revision, each DNA F2 individuals (Δ31) is used to detect the *ndrg1b* locus by PCR, followed by 10% acrylamide gel electrophoresis, step referred as 1^st^HMA, in which definitely would be possible to distinguish the bands derived from wild-type or KO fish (single band) according to their size. However, as Ansai and Kinoshita suggested in the book of Genome Editing in Animals – Methods and Protocols (Hatada, 2017), is recommended to perform a second HMA (2^nd^ HMA), for reliable distinction between wild-type and KO fish, following the next process: “to a PCR product showing a single band pattern in HMA, add a PCR product separately prepared from wild-type template. Reanneal the mixture by heating at 95 °C for 5 min and gently cooling to room temperature. Perform 2^nd^ HMA. The wild-type will show only one band, while KO fish will show a multiple banding pattern similar to heterozygotes”. Additionally, HMA method has been satisfactorily used in our laboratory to F2 screening for another locus (unpublished data). In Author response image 1 is an example of HMA for F2 screening: in the left panel is the 1^st^ HMA where heterozygous individuals were identified (+/- : 3, 4, 6, 7 and 8) because showed multiple banding patterns. In the right panel is the 2^nd^ HMA to clearly discriminate homozygous fish (individuals that showed a single band in 1^st^ HMA: 1, 2, 5, 9, 10, 11 and 12). Each PCR product from the 1st HMA was mixed with a wild-type template, to perform the 2^nd^ HMA, where showed wild-type fish (+/+ : 1, 5 and 10), and homozygous mutant (-/- : 9, 11 and 12). Fish 2 is discarded because its pattern is not clear.

**Author response image 1. sa2fig1:** 

References:Chai, A. W. Y., Yee, P. S., Price, S., Yee, S. M., Lee, H. M., Tiong, V. K. H., Gonçalves, E., Behan, F. M., Bateson, J., Gilbert, J., Tan, A. C., Mcdermott, U., Garnett, M. J., & Cheong, S. C. (2020). Genome-wide CRISPR screens of oral squamous cell carcinoma reveal fitness genes in the Hippo pathway. *eLife*, 9, e57761.

Delventhal, R., O’connor, R. M., Pantalia, M. M., Ulgherait, M., Kim, H. X., Basturk, M. K., Canman, J. C., & Shirasu-Hiza, M. (2019). Dissection of central clock function in *Drosophila* through cell-specific CRISPR-mediated clock gene disruption. *eLife*, 8, e48308.

Evers, B., Jastrzebski, K., Heijmans, J. P. M., Grernrum, W., Beijersbergen, R. L., & Bernards, R. (2016). CRISPR knockout screening outperforms shRNA and CRISPRi in identifying essential genes. Nature Biotechnology, 34(6), 631–633.

Fang, J., Chen, T., Pan, Q., Wang, Q., 2018. Generation of albino medaka (Oryzias latipes) by CRISPR/Cas9. J. Exp. Zool. Part B Mol. Dev. Evol. 330, 242–246. https://doi.org/10.1002/jez.b.22808

Feehan, J. M., Chiu, C. N., Stanar, P., Tam, B. M., Ahmed, S. N., & Moritz, O. L. (2017). Modeling Dominant and Recessive Forms of Retinitis Pigmentosa by Editing Three Rhodopsin-Encoding Genes in *Xenopus laevis* Using Crispr/Cas9. Scientific Reports, 7(1), 6920.

Foster, S. D., Glover, S. R., Turner, A. N., Chatti, K., & Challa, A. K. (2019). A mixing heteroduplex mobility assay (mHMA) to genotype homozygous mutants with small indels generated by CRISPR-Cas9 nucleases. MethodsX, 6, 1–5. https://doi.org/10.1016/j.mex.2018.11.017

Gandhi, S., Hutchins, E. J., Maruszko, K., Park, J. H., Thomson, M., & Bronner, M. E. (2020). Bimodal function of chromatin remodeler Hmga1 in neural crest induction and Wnt-dependent emigration. *eLife*, 9, e57779.

Hatada, I. (2017). Genome Editing in Animals. Methods and Protocols. In *Methods in Molecular Biology* (Vol. 1630, pp. 175–188). https://doi.org/10.1007/978-1-49397128-2

Gonçalves, D. M., & Oliveira, R. F. (2011). Hormones and Sexual Behavior of Teleost Fishes. In D. O. Norris & K. H. B. T.-H. and R. of V. Lopez (Eds.), Hormones and Reproduction of Vertebrates (pp. 119–147). Elsevier.

Hatada, I. (2017). Genome Editing in Animals. Methods and Protocols. In Methods in Molecular Biology (Vol. 1630, pp. 175–188). https://doi.org/10.1007/978-1-4939-7128-2

Homma, N., Harada, Y., Uchikawa, T., Kamei, Y., Fukamachi, S., 2017. Protanopia (red color-blindness) in medaka: a simple system for producing color-blind fish and testing their spectral sensitivity. BMC Genet. 18, 10. https://doi.org/10.1186/s12863-017-0477-7

Hockman, D., Chong-Morrison, V., Green, S. A., Gavriouchkina, D., Candido-Ferreira, I., Ling, I. T. C., Williams, R. M., Amemiya, C. T., Smith, J. J., Bronner, M. E., & Sauka-Spengler, T. (2019). A genome-wide assessment of the ancestral neural crest gene regulatory network. Nature Communications, 10(1), 4689.

Hoshijima, K., Jurynec, M. J., Klatt Shaw, D., Jacobi, A. M., Behlke, M. A., & Grunwald, D. J. (2019). Highly Efficient CRISPR-Cas9-Based Methods for Generating Deletion Mutations and F0 Embryos that Lack Gene Function in Zebrafish. Developmental Cell, 51(5), 645-657.e4.

Kayo, D., Zempo, B., Tomihara, S., Oka, Y., Kanda, S., 2019. Gene knockout analysis reveals essentiality of estrogen receptor β1 (Esr2a) for female reproduction in medaka. Sci. Rep. 9, 8868. https://doi.org/10.1038/s41598-019-45373-y

Kroll, F., Powell, G. T., Ghosh, M., Gestri, G., Antinucci, P., Hearn, T. J., Tunbak, H., Lim, S., Dennis, H. W., Fernandez, J. M., Whitmore, D., Dreosti, E., Wilson, S. W., Hoffman, E. J., & Rihel, J. (2021). A simple and effective F0 knockout method for rapid screening of behaviour and other complex phenotypes. *eLife*, 10, e59683.

Morgens, D. W., Deans, R. M., Li, A., & Bassik, M. C. (2016). Systematic comparison of CRISPR/Cas9 and RNAi screens for essential genes. Nature Biotechnology, 34(6), 634–636.

Naert, T., Colpaert, R., Van Nieuwenhuysen, T., Dimitrakopoulou, D., Leoen, J., Haustraete, J., Boel, A., Steyaert, W., Lepez, T., Deforce, D., Willaert, A., Creytens, D., & Vleminckx, K. (2016). CRISPR/Cas9 mediated knockout of rb1 and rbl1 leads to rapid and penetrant retinoblastoma development in *Xenopus tropicalis*. Scientific Reports, 6(1), 35264.

Ngamniyom, A., Magtoon, W., Nagahama, Y., & Sasayama, Y. (2009). Expression levels of hormone receptors and bone morphogenic protein in fins of medaka. Zoological Science, 26(1), 74–79.

Port, F., Strein, C., Stricker, M., Rauscher, B., Heigwer, F., Zhou, J., Beyersdörffer, C., Frei, J., Hess, A., Kern, K., Lange, L., Langner, N., Malamud, R., Pavlović, B., Rädecke, K., Schmitt, L., Voos, L., Valentini, E., & Boutros, M. (2020). A large-scale resource for tissue-specific CRISPR mutagenesis in *Drosophila*. *eLife*, 9, e53865.

Sanor, L. D., Flowers, G. P., & Crews, C. M. (2020). Multiplex CRISPR/Cas screen in regenerating haploid limbs of chimeric Axolotls. *ELife*, 9, e48511.

Sawamura, R., Osafune, N., Murakami, T., Furukawa, F., Kitano, T., 2017. Generation of biallelic F0 mutants in medaka using the CRISPR/Cas9 system. Genes to Cells 22, 756–763. https://doi.org/10.1111/gtc.12511

Seleit, A., Gross, K., Onistschenko, J., Woelk, M., Autorino, C., Centanin, L., 2020. Development and regeneration dynamics of the Medaka notochord. Dev. Biol. 463, 11–25. https://doi.org/https://doi.org/10.1016/j.ydbio.2020.03.001

Véron, N., Qu, Z., Kipen, P. A. S., Hirst, C. E., & Marcelle, C. (2015). CRISPR mediated somatic cell genome engineering in the chicken. Developmental Biology, 407(1), 68–74.

Williams, R. M., Senanayake, U., Artibani, M., Taylor, G., Wells, D., Ahmed, A. A., & Sauka-Spengler, T. (2018). Genome and epigenome engineering CRISPR toolkit for in vivo modulation of cis -regulatory interactions and gene expression in the chicken embryo. Development, 145(4), dev160333.

Yeh, Y.C., Kinoshita, M., Ng, T.H., Chang, Y.H., Maekawa, S., Chiang, Y.A., Aoki, T., Wang, H.C., 2017. Using CRISPR/Cas9-mediated gene editing to further explore growth and trade-off effects in myostatin-mutated F4 medaka (Oryzias latipes). Sci. Rep. 7, 1–13. https://doi.org/10.1038/s41598-017-09966-9